# Contribution to the Study of Lichenicolous Fungi from Northwest Iberian Peninsula (León and Lugo Provinces)

**DOI:** 10.3390/jof10010060

**Published:** 2024-01-12

**Authors:** Javier Etayo, María Eugenia López de Silanes

**Affiliations:** 1Navarro Villoslada 16, 3° dcha, 31003 Pamplona, Spain; 2Departamento de Enxeñaría dos Recursos Naturais e Medio Ambiente, Enxeñaría Forestal, Universidade de Vigo, Campus de Pontevedra, 36005 Pontevedra, Spain; esilanes@uvigo.gal

**Keywords:** fungi on lichens, new species, diversity, biogeography, taxonomy, Galicia, Spain, Europa

## Abstract

We have found 117 taxa of lichenicolous fungi in the studied area. In this paper, we describe five taxa: *Arthonia boomiana* on *Nephromopsis chlorophylla*, *Lawalreea burgaziana* on *Platismatia glauca*, *Pronectria scrobiculatae* on *Lobarina scrobiculata*, *Trichonectria parmeliellae* on *Parmeliella testacea* and *Trichonectria rubefaciens* ssp. *cryptoramalinae* on *Ramalina*. Furthermore, the next records are interesting chorologically from the Iberian Peninsula: *Arthophacopsis parmeliarum*, *Catillaria lobariicola*, *Lichenopuccinia poeltii*, *Myxotrichum bicolor*, *Nanostictis christiansenii*, *Niesslia lobariae*, *Opegrpaha sphaerophoricola*, *Pronectria fragmospora*, *Rhymbocarpus aggregatus*, *R. neglectus*, and *Tremella cetrariicola*.

## 1. Introduction

Lichenicolous fungi represent a successful group of organisms that live exclusively on lichens, most commonly as host-specific parasites but also as broad-spectrum pathogens, saprotrophs, and commensals [1]. However, we recorded some lichenized fungi and even a Myxomycete (Protista) living on lichens in the study area. These findings are not reflected in the title to ensure brevity.

The studied places are located in the Ancares and Courel mountain ranges, Galicia, and León, in the northwest of Spain. They belong to the European network Natura 2000 protected areas, decree 37/2014, and have an area of 102,634 ha. They have been designated by UNESCO as Biosphere Reserves (http://rerb.oapn.es/, accessed on 26 October 2023). The valleys in these mountains are between hillsides with steep slopes; in Ancares, the maximum altitude is that at Pico Miravalles, of 1969 m, whereas Courel is lower, with a maximum altitude of 1639 m. These two zones host some of the best-preserved Atlantic forests in the NW of the Iberian Peninsula. In Ancares, the forest formations are altimontane and are dominated by *Betula pubescens*, montane *Quercus pyrenaica*, oligotrophic *Alnus glutinosa*, and *Salix atrocinerea*. In addition to these trees, Courel is also characterized by a combination of *Fagus sylvatica* and *Quercus ilex* forests. Both places are characterized by large areas occupied by ancient cultivated *Castanea sativa*, many of which are in a state of abandonment, but they have a high level of lichen biodiversity [2].

Neither in Galicia nor in these mountains have any studies been previously carried out exclusively on lichenicolous fungi, despite several studies describing lichen biodiversity in which lichenicolous are occasionally recorded [3,4,5,6,7,8]. A recent study by van den Boom et al. [9] described a new species of lichenicolous *Micarea* from Courel.

## 2. Materials and Methods

A magnifying stereoscope (LEICA S9i with built-in cameras LASX and NIKON S47U, Tokyo, Japan) and a microscope (NIKON ECLIPSE 80i with a Progres CT1 camera, Tokyo, Japan) were used for morphological and anatomical analysis of the specimens. The measurements of structures that could be observed using the magnifying glass, such as apothecia, were carried out dry. For microstructures, measures were made in water. The number of measures is only indicated for the newly described species. Sections of the ascomata were made with a freehand razor blade. Conventional reagents in lichenological studies were used: potassium hydroxide, 10% (K) Lugol’s solution (I) isolated or after pretreatment with K (KI) nitric acid (N) and cresyl blue (BCr) or Congo red (CR); the latter two were used to color certain fungal structures. The nomenclature used to classify the fungi was that of Diederich et al. [1].

The specimens are stored in the private herbarium of the first author (hb. Etayo) and the herbarium of the Faculty of Pharmacy, Botany Laboratory, University of Santiago de Compostela, Spain (SANT-Lich.).

### Localities List

Loc. 0. Lugo, Pedrafita do Cebreiro, rest area LU-623, 42°43′30.4″ N, 7°01′23.2″ W, 1115 m, 4–July–2021, J. Etayo and M.E. López de Silanes.

Loc. 1. Lugo, Ancares, Cabaniños, between Piornedo and Campa da braña, old *Castanea sativa*, 42°50′10.9″ N, 6°54′06.2″ W, 960–970 m, 2–July–2021, J. Etayo and M.E. López de Silanes.

Loc. 2a. Lugo, Ancares, path towards Cabana Vella, treeless area, slope, 42°48′45.2″ N, 6°54′27.8″ W, 1350–1400 m, 2–July–2021, J. Etayo and M.E. López de Silanes.

Loc. 2b. Lugo, Ancares, path towards Cabana Vella, forest area near Cabana Vella, 42°48′57.6″ N, 6°53′56.3″ W, 1160 m, 2–July–2021, J. Etayo and M.E. López de Silanes.

Loc. 3a. León, Ancares, Teso de Valiñas, road LE-3203, *Quercus petraea* forest, granitic rocks, 42°51′53.1″ N, 6°51′47.1″ W, 1155–1165 m, 3–July–2021, J. Etayo and M.E. López de Silanes.

Loc. 3b. León, Ancares, Teso de Valiñas, road LE-3203, *Quercus petraea*, *Betula pubencens* near the river, granitic rocks, 42°51′55.1″ N 6°51′38.4″ W, 1084 m, 3–July–2022, J. Etayo and M.E. López de Silanes.

Loc. 4. León, Ancares, Puerto de Ancares, Mirador de Balouta, stone slate, 42°52′07.0″ N, 6°49′05.5″ W, 1650 m, 3–July–2021, J. Etayo and M.E. López de Silanes.

Loc. 5. Lugo, Ancares, Piornedo, wooden fences and stone walls of granitic rocks 42°51′26.1″ N, 6°52′35.6″ W, 1130 m, 4–July–2021, J. Etayo and M.E. López de Silanes.

Loc. 6. Lugo, rocky slope on road border P–LU-3505, 42°52′53″ N, 6°55′36″ W, 730 m, 4–July–2021, J. Etayo and M.E. López de Silanes.

Loc. 7. Lugo, Silvouta, road P-LU 3505, *Prunus avium*, *Castanea sativa*, 42°54′52.6″ N, 6°59′18.2″ W, 789 m, 4–July–2021, J. Etayo and M.E. López de Silanes.

Loc. 8. Lugo, Faial de Liñares, Liñares LU-623, near peak of San Roque, *Fagus sylvatica* forest, 42°41′58.3″ N, 7°04′51.4″ W, 1100–1200 m, 4–July–2021, J. Etayo and M.E. López de Silanes.

Loc. 9. Lugo, Courel, Devesa da Rogueira, *Betula pubescens*, *Quercus pyrenaica*, *Fagus sylvatica*, 42°36′29.6″ N, 7°06′21.4″ W, 1200–1300 m, 5–July–2021, J. Etayo and M.E. López de Silanes.

Loc. 10. Lugo, Courel, Devesa da Rogueira, lower zone after stream, descending on right side, oak, and calcareous and siliceous slopes, 42°37′23.6″ N, 7°06′52.5″ W, 750–900 m, 5–July–2021, J. Etayo and M.E. López de Silanes.

Loc. 11. Lugo, Courel, Mazo village, from Seoane to Paderme on riverside of Pequeno River, 42°38′58.9″ N, 7°09′35.1″ W, 620–650 m, 6–July–2021, J. Etayo and M.E. López de Silanes.

Loc. 12. Lugo, Courel, Parada village, *Castanea sativa* managed forest with old chestnuts, 42°37′30.1″ N, 7°08′09.4″ W, 700 m, 6–July–2021, J. Etayo and M.E. López de Silanes.

Loc. 13. Lugo, Courel, Mercurín, *Castanea sativa* unmanaged forest with old chestnuts, 42°37′58″ N, 7°09′48.6″ W, 670 m, 6–July–2021, J. Etayo and M.E. López de Silanes.

## 3. Results

The list of identified taxa is presented in alphabetical order. For each, we describe the locality where it was found, the substrate, and the herbaria where the specimens are deposited. Five taxa are described as new to science, and differences with related species are discussed. Differential data are also provided for those taxa we think could require some clarification.

### Catalogue List

***Abrothallus bertianus*** De Not.

It is quite common, especially in its asexual form (*Vouauxiomyces*).

Loc. 0, on *Melanohalea exasperata* on twig of *Acer* sp., J. Etayo 33,461 (SANT-Lich. 12625). Loc. 2b, on *Melanelixia subaurifera* on *Quercus* sp., J. Etayo 33,577 (SANT-Lich. 12578). Loc. 2b, on *M. exasperata* on *Quercus* sp., J. Etayo 33,580 (hb. Etayo). Loc. 8, on *Melanelixia glabratula* on *Ilex aquifolium* and *Fagus sylvatica*, J. Etayo 33,357 (hb. Etayo). Loc. 9, on *M. exasperata* on twigs of a tree, J. Etayo 33,632 (hb. Etayo).

***Abrothallus lobariae*** (Diederich & Etayo) Diederich & Ertz

≡ *Phoma lobariae* Diederich & Etayo

Described in Etayo and Diederich [10]. It has recently been considered the asexual morph of an *Abrothallus* and named *A. lobariae* [1]

Loc. 3a, on *Lobaria pulmonaria* growing on *Quercus petraea*, J. Etayo 33,410 (hb. Etayo). Loc. 9, on *L. pulmonaria* growing on *Fagus*, J. Etayo 33,628 (hb. Etayo). Loc. 12, on *L. pulmonaria* growing on *Castanea*, J. Etayo 33,565 (SANT-Lich. 12607). Ibidem, M.E. López de Silanes (SANT-Lich. 12697).

***Abrothallus parmeliarum*** (Sommerf.) Arnold

Apparently quite rare, it is found as a few apothecia on the lobes of *Parmelia*, coexisting with *Rinodina efflorescens* and *Sphaerellothecium parmeliae.*

Loc. 3a, on *Parmelia saxatilis* growing on *Quercus petraea*, J. Etayo 33,431 (hb. Etayo). Ibidem, on *P. saxatilis* on *Q. petraea*, M.E. López de Silanes (SANT-Lich. 12482).

***Abrothallus suecicus*** (Kirstch.) Nordin

Characterized by its three septate spores and found on *Ramalina calicaris* and *R. farinacea*. This species seems to be quite rare in the area.

Loc. 1, on *Ramalina calicaris* and *R. farinacea* growing on *Castanea sativa*, J. Etayo 33,381 (hb. Etayo).

***Abrothallus usneae*** Rabenh.

The violaceous hymenium characterizes this taxon [11], whose apothecia appear dispersed on branches of *Usnea.* On different areas of host thallus, there are brown round spots, maybe a prelude of this species. We also found it on the basidiomata of *Biatoropsis*, sometimes almost covering it completely.

Loc. 1, on blackened thallus of *Usnea* sp. on *Castanea sativa*, J. Etayo 33,370 (hb. Etayo). Ibidem, on *Biatoropsis* on *Usnea* sp., J. Etayo 33,385 (hb. Etayo). Loc. 2b, on *Usnea* sp. on *Corylus*, J. Etayo 33,587 (SANT-Lich. 12579). Ibidem, J. Etayo 33,589 (hb. Etayo). Loc. 3a, on *Biatoropsis on Usnea* sp. on *Quercus petraea*, J. Etayo 33,414 (hb. Etayo). Loc. 2b, on *Usnea* sp. on *Quercus*, J. Etayo 33,576 (hb. Etayo).

***Acremonium*** sp.

Conidiogenous cells, 40–120 µm, hyaline, septate, and apically provided with bacillar to ellipsoid conidia, 5–6 × 1.5–2 µm. It was found on the apex of the laciniae of *Evernia prunastri* mixed with *Lichenoconium erodens* and on thalli of *Flavoparmelia caperata*.

Loc. 7, on *Evernia prunastri* growing on *Betula*, J. Etayo 33,468 (hb. Etayo). Ibidem, on brownish soralia of *Flavoparmelia caperata* growing on *Betula*, J. Etayo 33,470 (hb. Etayo).

***Actinocladium rhodosporum*** Ehrenb.

A facultatively lichenicolous fungus that has been recorded on several host genera: *Flavoparmelia*, *Hypotrachyna*, *Lecanora*, *Lepra*, and *Lepraria* [12], and especially *Usnea* in Navarra (unpublished data from J. Etayo). Here, we report it on *Lepra albescens*. In Spain, it was known as lichenicolous from Basque Country [13].

Loc. 3a, on *Lepra albescens* growing on *Quercus petraea*, J. Etayo 33,456 (hb. Etayo).

***Arthonia boomiana*** Etayo, sp. nov. (Figure 1)

MycoBank MB 851490

Differs from *A. xanthoparmeliarum* in the larger apothecia, 150–500 µm diam. vs. 100–150(–200) µm diam. in *A. xanthoparmeliarum*, dark brown vs. light brown hypothecium, hardly colored paraphysoids, only widened in the apex vs. strongly capitate and colored upwards, non halonate vs. halonate ascospores and the host selection: growing on *Nephromopsis* vs. on *Xanthoparmelia.*

TYPUS. SPAIN, León, Ancares, Teso de Valiñas, road LE-3203, *Quercus petraea* forest, granite rocks, on *Nephromopsis chlorophylla* growing on *Q. petraea*, 42°51′53.1″ N 6°51′47.1″ W, 1155–1165 m, 3–July–2021, J. Etayo 33,444 and M.E. López de Silanes (SANT-Lich. 12713–holotype designated here, hb. Etayo–isotype).

Ascomata apothecioid, lichenicolous on the lobules, especially near their border, of *N. chlorophylla*, superficial, black, matt, convex since young to almost subspherical, 0.15–0.50 mm diam., sometimes aggregated, containing 12–15 apothecia, then forming groups up to 1 mm diam., always surrounded by a necrotic, brown, 200–300 µm wide rim. Epihymenium brownish, K+ olivaceous green or brownish. Hypothecium dark brown, K+ olivaceous brown or olivaceous black, to 80 µm thick in its central part, of textura angularis with small cells c. 4–6 µm diam., with irregularly thickened wall. Hymenium colorless, I+ blue, KI+ blue, 30–40 µm tall. Paraphysoids not distinguishable between the asci but forming a net of roundish cells 3–4(–5) µm diam. above them. Asci widely clavate, 8-spored, KI-, with a wide apical dome but without a KI+ blue ring around ocular chamber, 29–37 × 14–17 µm (n = 8). Ascospores first colorless and smooth-walled, brown and verruculose when overmature, ellipsoid, straight, 1-septate, not constricted at the septum, with many small oil droplets inside, (12–)13–15(–16) × 4.5–5.5 µm (n = 26).

Etymology: *Arthonia boomiana* is dedicated to Pieter P.G. van den Boom, a good friend and a very experienced lichenologist who has shared much information and samples with the first author during the recent decades.

No other species of *Arthonia* has been known on *Nephromopsis*, not even on other genera related to *Cetraria* [14]. Its habitus looks like *Arthonia epiphyscia.* However, in this species, ascomata are smaller, up to 350 µm diam., hymenium I+ red, and smaller ascospores that are always hyaline, 10–14 × 3.5–5 µm [15]. Another species growing preferentially on the thallus border is *A. marginalis*, but the host of this species is *Pseudocyphellaria freycinetii*, its ascomata are smaller, 50–120 µm diam., apothecial pigment is green, and ascospores are also smaller, 10.5–13 × 4–5.2 µm [16]. *Nephromopsis chlorophylla* is a genus of Parmeliaceae [15], so we should compare *Arthonia boomiana* with other species growing on *Parmelia* s. lat. From them, we think the most similar in habitus are *A. pepei* and *A. xanthoparmeliarum*. *A. pepei* differs by its smaller apothecia, 70–140 µm, growing on a pinkish to cream-colored infection, colorless subhymenium, blackish epihymenium, asci with a clear apical ring (in KI), smaller ascospores (8–)9.5–11 × 4–4.5 μm, and a different host, *Parmelina cryptotiliaceae* [17]. *Arthonia xanthoparmeliarum* differs in its smaller apothecia 100–150(–200) µm, light-brown hypothecium, strongly capitate and colored upward paraphysoids, ascospores with a perispore, and the different host selection, *Xanthoparmelia* [16].

It is only known from the type locality that it was living with *Tremella coppinsii* on thalli of corticolous *Nephromopsis chlorophylla*.

***Arthonia coronata*** Etayo

We found some excellent populations of this species in the region. Its apothecia grow directly on the thallus of *Flavoparmelia caperata* on small discolored infections (beige or cream brown) surrounded by a black necrotic area, and they have big ascomata, where hairs can be seen even through a magnifying glass.

Loc. 11, on soralia of *Flavoparmelia caperata* on *Castanea sativa*, J. Etayo 33,485 (hb. Etayo). 13, on *F. caperata* on *C. sativa*, J. Etayo 33,547 (hb. Etayo). Ibidem on *F. caperata* on *C. sativa*, M.E. López de Silanes (SANT-Lich 12566).

***Arthonia digitata*** Hafellner

It was found on the upper parts of squamules of *Cladonia digitata*, usually forming groups where ascomata sometimes overlap with conidiomata of *Milospium lacoizquetae.*

Loc. 1, on squamules and podetia of *C. digitata* growing on wood of *Castanea sativa*, J. Etayo 33,384 (hb. Etayo). Loc. 11, on *C. digitata* growing on wood of *C. sativa*, J. Etayo 33,478 (hb. Etayo). Loc. 13, on squamules of *C.* cf. *digitata*, on *C. sativa*, M.E. López de Silanes (SANT-Lich 12711)

***Arthonia excentrica*** Th. Fr.

Our taxon has dark brown epithecium and hypothecium; hymenium c. 35 µm high, I+ red; asci 8-spored and ascospores 1-septate, constricted at the septum and with several oil droplets inside, 8.5–12.5 × 4–5 µm, a little thinner than in Ihlen and Wedin [18]

Loc. 11, on *Lepraria lobificans* growing on *Castanea sativa*, J. Etayo 33,490 (SANT-Lich., hb. Etayo).

***Arthonia parietinaria*** Hafellner & A. Fleischhacker

Loc. 7, on *Xanthoria parietina* growing on *Sambucus nigra*, J. Etayo 33,471 (hb. Etayo).

***Arthophacopsis parmeliarum*** Hafellner

We rarely found it forming convex galls scattered on *Parmelia sulcata*.

Loc. 2b, on *Parmelia sulcata* growing on *Corylus*, J. Etayo 33,592 (hb. Etayo). Loc. 8, on *P. sulcata* growing on *Fagus sylvatica*, J. Etayo 33,367 (hb. Etayo). Loc. 9, on *P. sulcata* growing on *Fagus*, J. Etayo 33,636 (hb. Etayo).

***Athelia arachnoidea*** (Berk.) Jülich

We found thalli of *Lepra albescens* and *Platismatia* with brown sclerotia and arachnoid mycelium typical of *Athelia.*

Loc. 9, on *Lepra albescens* growing on *Fagus sylvatica*, J. Etayo 33,624 (hb. Etayo). Ibidem, on *Platismatia glauca* growing on *F. sylvatica*, J. Etayo 33,640 (hb. Etayo).

***Biatoropsis usnearum*** Räsänen s. lat.

Most basidiomata are erumpent along the thallus, concave, flat even tuberculate, and dark brown or black in color.

Loc. 1, on branches of *Usnea florida* growing on *Castanea sativa*, J. Etayo 33382, 33,385 (hb. Etayo). Loc. 2a, on *Usnea* sp. growing on bushes, J. Etayo 33,675 (hb. Etayo). Loc. 2b, on *Usnea* sp. growing on *Quercus* sp., J. Etayo 33,576 (hb. Etayo). Loc. 3a, abundant on *Usnea* sp. growing on *Q. petraea*, J. Etayo 33,414 (hb. Etayo). Loc. 8, on *Usnea* gr. *florida* growing on *Fagus sylvatica*, J. Etayo 33,353 (hb. Etayo). Loc. 9, on *Usnea* sp. growing on *F. sylvatica*, J. Etayo 33,637 (hb. Etayo, SANT-Lich. 12610).

***Capronia*** sp.

We found it seldomly growing on black spots on unhealthy thalli of *Parmelia sulcata* accompanied by a hyphomycete with brown, vertical, long, septated conidiophores and small, grey, and simple conidia. Perithecia of *Capronia* are sessile, c. 100 µm diam., with abundant hairs all around that are thin, 20–50 × 3–4 µm, and bulbiform basal part 4–6(–8) µm, 0–2-septate, straight to more habitually curved, with acute ends. We did not find asci or spores inside.

Loc. 8, on *Parmelia sulcata* growing on *Fagus sylvatica*, J. Etayo 33,367 (hb. Etayo).

***Carbonea vitellinaria*** (Nyl.) Hertel

Loc. 4, on *Candelariella vitellina* growing on exposed rocks, J. Etayo 33,600 (hb. Etayo).

***Catillaria* cf. *lobariicola*** (Alstrup) Coppins & Aptroot

We found dispersed apothecia growing on the upper side of *Lobarina scrobiculata* lobes, which are similar to those of *C. lobariicola* known from Scotland and Norway [15,19] and from Portugal [20]. We also have another record from Navarra.

Apothecia are cream- to brown-colored, not black as in original description. Exciple, brown in the original description [19], is different on our samples, being hyaline, formed by a cartilaginous tissue, and only brown on the outer part. In one sample, we found only dispersed pycnidia and hyaline conidia with obtuse ends, 5–7 × 2–2.5 µm, with two hyaline droplets.

Loc. 10, on *Lobarina scrobiculata* on *Fagus sylvatica*, J. Etayo 33,667 (hb. Etayo). Loc. 3a, on *L. scrobiculata* on *Quercus petraea*, J. Etayo 33,412 (hb. Etayo).

Another studied locality: Spain, Navarra, valle de Arce, between Oroz-Betelu and Olaldea, on *L. scrobiculata* on *Quercus* forest, 630 m, 42°54′48″ N, 1°17′50″ W, 10–September–2017, J. Etayo 30,805 (hb. Etayo).

***Catillaria nigroclavata*** (Nyl.) Schuler

This species has been found frequently on corticolous lichens [17,21]. Hafellner [22] found it, like we did, on species of *Parmelina*.

Loc. 12, on *Parmelina pastillifera* on twig of *Juglans*, J. Etayo 33,555 (SANT-Lich. 12609).

Other studied Spanish localities growing on *Parmelina* hosts: Tarragona, Poblet, Paraje Natural de Poblet, path from information center toward Mirador de La Pena, 41°21′59″ N, 1°04′53″ W, 700–800 m, 14–July–2020, on *P. quercina* growing on *Quercus ilex*, J. Etayo 32,484 (hb. Etayo). Navarra, Belascoain, in front of a watering place, mixed forest on a slope, 42°45′32″ N, 1°50′15″ W, 380 m, 10–04–2016, on *P. quercina* growing on *Acer*, J. Etayo s.n. (hb. Etayo). La Rioja, Tierra de Cameros, Gallinero de Cameros, forest along a stream, 1000 m, 11–April–1996, on *P. quercina* growing on *Quercus* sp., J. Etayo 14,286 (hb. Etayo).

***Cecidonia umbonella*** (Nyl.) Triebel & Rambold

Loc. 4, on *Lecidea swartzioidea* on exposed rocks, J. Etayo 33,601 (hb. Etayo).

***Chalara lobariae*** Etayo

Loc. 9, on unhealthy thalli of *Lobaria pulmonaria* growing on *Fagus sylvatica*, J. Etayo 33,627 (hb. Etayo). Ibidem, on *L. pulmonaria* growing on *F. sylvatica*, J. Etayo 33,642 (hb. Etayo). Loc. 10, on *L. pulmonaria* growing on *F. sylvatica*, J. Etayo 33,656 (SANT-Lich. 12616).

***Cladophialophora parmeliae*** (Etayo & Diederich) Diederich & Unter.

A common species that can be found on different lichens. In the studied area, we found it mostly on the apothecia of *Pectenia plumbea*.

Loc. 3a, on *P. plumbea* growing on *Quercus petraea*, J. Etayo 33,420 (hb. Etayo).

***Crittendenia coppinsii*** (P. Roberts) Diederich, M. Westb., Millanes & Wedin

This species, only known from Europe (Austria, Belgium, Norway, Russia, Sweden, and Switzerland, [23]) was here found in Courel (Lugo), an important area, where it was very abundant. Many thalli of *Melanelixia glabratula* were covered by its small basidiomata.

Loc. 8, on *M. glabratula* growing on *Corylus avellana*, J. Etayo 33,361 (hb. Etayo). Loc. 9, on *M. glabratula* growing on *Fagus sylvatica*, J. Etayo 33,615 (hb. Etayo). Ibidem M.E. López de Silanes (SANT-Lich. 12701)

***Dacampia rufescentis*** (Vouaux) D. Hawksw.

Loc. 10, on *Peltigera rufescens* on soil, J. Etayo 33,650 (hb. Etayo, SANT-Lich. 12617).

***Didymocyrtis foliaceiphila*** (Diederich, Kocourk. & Etayo) Ertz & Diederich

It has been identified on the Iberian Peninsula [24,25] on some species of *Cladonia* and *Parmelia sulcata*. One of our samples was growing on soralia of *Hypogymnia tubulosa*, a host not recorded in the literature.

Loc. 1, on squamules and podetia of *Cladonia* cf. *digitata* on wood of *Castanea sativa*, J. Etayo 33,369 (hb. Etayo). Ibidem, on soralia of *Hypogymnia tubulosa* growing on *C. sativa*, J. Etayo 33,391 (hb. Etayo).

***Didymocyrtis pseudeverniae*** (Etayo & Diederich) Ertz & Diederich

It is a strict parasite of *Pseudevernia furfuracea*, easily recognizable because infected thalli become pink grey, pycnidia remain fully immersed, and conidia are very big.

Sometimes, we found infected thalli most likely with its teleomorph, not described so far. As we did not perform molecular studies, their determination is not conclusive. We did not find them on pink spots like the anamorph, but on yellowish- or brown-colored thalli; perithecia were completely immersed in the thallus. Ascomatal wall is composed of several layers, outer layer formed by polygonal, subspherical or ellipsoid cells with a thin wall, except on the angles, where it is thicker, darker and with extracellular pigment. Pseudoparaphyses are septate, not capitated, 2–3 µm wide, simple to occasionally branched over the asci. Asci bitunicate, cylindrical, apically widened, 8-spored, 100–108 × 9.5–11 µm. Ascospores 1-septate, hyaline when young, brown with a verruculose wall when mature, surrounded by a gelatinous sheath 1–2 µm wide, strongly constricted at the septum, with obtuse to more frequently subacute apices too, 14–17.5 × 7–9 µm (22).

Loc. 3b, on *Pseudevernia furfuracea* growing on *Quercus petraea* and *Betula pubescens*, J. Etayo 33,425 (hb. Etayo). Loc. 2b (teleomorph) on *P. furfuracea* growing on *Corylus avellana*, J. Etayo 33,590 (hb. Etayo). Loc. 9, on *P. furfuracea* growing on *Quercus pyrenaica*, J. Etayo 33,618 (hb. Etayo).

***Didymocyrtis ramalinae*** (Roberge ex Desm.) Ertz, Diederich & Hafellner

This is a common species in the studied zone, especially on *Ramalina fastigiata*, but not on the nearby thalli of *R. farinacea.* Interestingly, it also appears on *Evernia prunastri*, something recorded previously [26].

Loc. 1, on *Ramalina fastigiata*, growing on *Castanea sativa*, J. Etayo 33,380 (hb. Etayo). Loc. 7, on *Evernia prunastri* growing on *Prunus cerasus*, J. Etayo 33,475 (SANT-Lich. 12627). Loc. 11, on *Ramalina calicaris* growing on *C. sativa*, J. Etayo 33,484 (hb. Etayo).13, on *E. prunastri* growing on *C. sativa*, J. Etayo 33,546 (SANT-Lich. 12624).

***Diploschistes muscorum*** (Scop.) R. Sant.

Loc. 10, on *Cladonia* growing in a chalk crevice, J. Etayo 33,652 (hb. Etayo).

***Endococcus alpestris*** D. Hawksw.

We found this species on blackened parts of main branches of *Usnea florida* and *Usnea* sp. Some features of it are ascomata small, immersed or semi-immersed with brown wall, K+ greenish brown; asci without foot, 35–40 × 7–9 µm and ascospores uni- to biseriate, 8–11.5 × 4–5 µm. One of the samples on *Usnea* sp. had ascospores easily broken at the septum.

We found also a sample growing on *Ramalina* with a little smaller and not sessile perithecia. *Endococcus ramalinarius* was known from New Zealand on *Ramalina leioidea*, and it has wider asci, 29–35 × 9–14 µm, and ascospores, 10–13 × 4–5 µm, slightly larger than those of our specimen on *Ramalina*.

Loc. 1, on blackened thallus of *Ramalina farinacea* growing on *Castanea sativa*, J. Etayo 33,379 (hb. Etayo). Ibidem, on *Usnea florida* growing on *C. sativa*, J. Etayo 33,402 (hb. Etayo). Loc. 11, on *Usnea* sp. growing on *C. sativa*, J. Etayo 33,492 (SANT-Lich. 12601).

***Endococcus*** aff. ***parmeliarum*** Etayo

This species was described growing on *Parmelia* (and also probably on *Usnea*) from Chile [16]. Our sample on *Nephromopsis* has brown-greenish ascomata, asci bitunicate, hamathecium present in young material, filaments composed of moniliform cells, soon disappearing. Ascospores are a little longer than in the type, 10.5–11 × 5.5 µm, and with a thick gelatinous sheath of 2–3 µm.

Loc. 7, on *Nephromopsis chlorophylla* growing on *Betula*, J. Etayo 33,469 (hb. Etayo).

***Endococcus rugulosus*** (Nyl. ex Malbr.) Arnold

This species is quite common on *Aspicilia* in this locality.

Loc. 4, on *Aspicilia caesiocinerea* growing on exposed rocks, J. Etayo 33,599 (hb. Etayo).

***Endococcus umbilicariae*** (Linds.) Hafellner

≡ Polycoccum umbilicariae (Linds.) D. Hawksw.

Recently combined in *Endococcus*, this apparently rare species was known to be growing only on species of *Lasallia* (now synonym of *Umbilicaria*) from Europe, Canary Islands, and Madeira. In mainland Spain, it was known from Madrid [27] and Cáceres [28].

Loc. 4, on *Umbilicaria hispanica* on exposed rocks, J. Etayo 33,602 (hb. Etayo).

***Epicladonia sandstedei*** (Zopf) D. Hawksw.

Loc. 3a, on *Cladonia* cf. *subcervicornis* growing on soil, J. Etayo 33,439 (hb. Etayo).

***Epicladonia stenospora*** (Harm.) D. Hawksw.

Loc. 11, on *Cladonia* cf. *coniocraea* growing on *Castanea sativa*, J. Etayo 33,483 (hb. Etayo). Loc. 12, on *Cladonia* sp. growing on *C. sativa*, J. Etayo 33,573 (hb. Etayo).

***Epithamnolia xanthoriae*** (Brackel) Diederich & Suija

Conidiomatal wall with long hyphae of textura porrecta and septate; simple or branched conidiophores are typical of the genus. This species has been recorded on a large number of hosts [29] but never on *Usnea*. The conidiomata in our sample were growing on circular, black dots. Conidia are straight or curved, 3–5-septate, 50–52 × 2–2.5 µm (when mature).

Loc. 1, on *Usnea florida* growing on *Castanea sativa*, J. Etayo 33,385 (hb. Etayo). Loc. 3a, on *Usnea* sp. growing on *Quercus petraea*, J. Etayo 33,447 (hb. Etayo).

***Feltgeniomyces dichotomus*** Etayo & Calat.

Loc. 2b, on *Physcia aipolia* growing on *Quercus* sp., J. Etayo 33,581 (hb. Etayo).

***Hemigrapha*** cf. ***asteriscus*** (Müll. Arg.) D. Hawksw.

Ascomata not seen, conidiomata have the same structure as thyriothecia, flattened but with a very uneven surface, black, at first round then star-shaped, 50–150(–200) µm diam., composed of radiating rows of polyhedral cells, 3–5 × 2–3.5 µm diam. Conidiogenous cells ampulliform, 4–5 × 4 µm, conidia ellipsoid, hyaline, with obtuse apex, basally a bit truncate, with small oil droplets at the ends, (3.5–)4–4.5(–5) × 1.5–2 µm.

Species of *Hemigrapha* growing on Peltigerales were studied by Diederich and Wedin [30]. These species have two known asexual stages: one producing macroconidia, previously known from *H. asteriscus* and *H. pseudocyphellariae*, and one producing only microconidia, known from *H. asteriscus* and *H. atlantica* [30]. Since the sizes of the conidiomata and conidia in our specimen are very similar to microconidial anamorph of *H. asteriscus*, we believe that this species so far only collected on *Peltigera* species could also colonize *Lobaria pulmonaria*. Furthermore, the only sample of *H. asteriscus* known from Europe (Switzerland) in Diederich and Wedin [30] was also only with conidiomata.

Loc. 12, on unhealthy whitened *Lobaria pulmonaria* growing on *Castanea sativa*, J. Etayo 33,564 (hb. Etayo).

***Intralichen*** cf. ***baccisporus*** D. Hawksw. & M.S. Cole

This species was described by Hawksworth and Cole [31] on the hymenium of *Caloplaca trachyphylla* in the USA (Nebraska), who noted that it has intermediate characteristics between *I*. *christiansenii* and *I. lichenicola.* Our sample lives in the hymenium of *Lecanora chlarotera* growing on *Castanea sativa*, which turns black under infection and has conidia, 2–11 cells, 8.5–16 × 6–8 µm.

Loc. 12, on *Lecanora chlarotera* growing on *Castanea sativa*, J. Etayo 33,567 (hb. Etayo).

***Lambiella insularis*** (Nyl.) T. Sprib.

≡ *Rimularia insularis* (Nyl.) Rambold & Hertel

Loc. 4, on *Lecanora rupicola* growing on exposed boulder, J. Etayo 33,608 (hb. Etayo).

***Lawalreea burgaziana*** Etayo & López de Silanes, sp. nov. (Figure 2)

MycoBank MB 851491

Differs from *Lawalreea lecanorae* in having larger conidiomata, 30–100 µm diam. vs. 15–25 µm, larger conidiogenous cells and conidia these last are curved, (5–)5.5–6.5(–7.5) × 2–3 µm vs. straight, 2.2–4.5 × 1.2–1.5 µm, and the host selection, *Platismatia* vs. *Lecanora*.

TYPUS. SPAIN, Lugo, Courel, Devesa da Rogueira, lower zone after stream, descending on right side, oak and calcareous and siliceous slopes, on *Platismatia glauca* growing on *Fagus*, 42°37′23.6″ N 7°06′52.5″ W, 750–900 m, 5–July–2021, J. Etayo 33,663 and M.E. López de Silanes (SANT–Lich. 12714 holotype designated here, hb. Etayo– isotype).

Distinct vegetative hyphae not observed. Conidiomata very abundant on the upper surface of the host, sessile, dispersed but settled at a regular distance, pycnidia unilocular, blackish green, ostiolate, ostiole width with an opening of 7–20 µm diam., finally becoming similar to an acervulum, 15–25 µm diam. Conidiomatal wall light green to green bluish above K-, N+ brown violaceous, hyaline below, 4–6 µm thick, with textura angularis in surface view, cells isodiametric with thin wall, 2–4 µm diam. Hymenium KI-. Conidiophores reduced to conidiogenous cells. Conidiogenesis enteroblastic, phialidic. Conidiogenous cells lining the conidiomatal cavity, hyaline, mostly nonproliferating, rarely percurrent, without or with one visible annellation, ampulliform to subglobose, sometimes with a thin beak, 3–4 × 2.5–3 µm. Conidia colorless, dry, with a smooth and thin wall, without guttules or with two small ones in the extremes, simple, thinly ellipsoid to bacillar, straight, with rounded apex and a slightly truncated base, 2.2–4.5 × 1.2–1.5 µm. Sexual morph not observed.

*Viridiannula pertusariae* [32] seems to be similar. Habitus of its conidiomata are similar in color and form but slightly larger, 40–50(–70) µm diam. Conidiogenous cells are, however, different, polyphialidic, with 1–4 proliferations, and larger, 5–11 × 2–4(–5); conidia are also larger and more polymorphic, 6.5–10(–12) × 2.5–3(–3.5) µm. *Diplolaeviopsis* [33] also has greenish wall but differs in its holoblastic conidiogenesis, and then conidia have conidiogenous cells resting in the base that are also 1-septate. By its extremely small size of conidiomata, *Lawalreea burgaziana* could be compared with *Minutophoma* [34], but that genus does not have a differentiated wall below the pycnidia and the conidiogenous cells occupy that basal zone. Furthermore, conidiomatal wall is dark brown. *Briancoppinsia*, with the species *B. cytospora* [≡ *Phoma cytospora*] grows predominantly on parmelioid lichens (including *Platismatia glauca*) but also on *Cladonia* sp., *Lecanora conizaeoides*, *Lepra* sp., etc. [12,35]. *Briancoppinsia cytospora*, however, has many differences with *L. burgaziana*: conidiomata are typically pycnidia, closed, firstly immersed then erumpent breaking the thallus, much larger, (40–)48–81(–120) μm diam.; hymenium is KI+ blue; pycnidial wall is composed of compacted entwined short-celled hyphae forming a textura intricata and conidiogenous cells and conidia are also larger, these last being (4.5–)5.8–6.8(–8.0) × (1.2–)1.6–2.0(–2.6) μm and slightly curved [35]. These authors [12,35] also explained the reasons they changed the lost type (on *Flavoparmelia caperata*) for a sample growing on *Lecanora expallens*. In our opinion, this was not the best decision because infections on Parmeliaceae and on crustaceous lichens are different. On Parmeliaceae, infection causes a brownish necrotic areas surrounded by a black rim while in *Lecanora conizaeoides*, infection is hardly visible. As *Briancoppinsia cytospora* was recorded growing on *Platismatia*, we thought our fungus could be a microconidial morph of *Briancoppinsia*. However, color as well as structure of the conidiomatal wall is different. Finally, we assumed that those extremely small structures could be pycnidia of the host, but, according to Ryan [36], conidiomata in *Platismatia* are marginal and with bacillar conidia, 4–7 × 1 µm.

The monotypic genus *Lawalreea* was described [37] for a fungus with pycnidia that open irregularly at the apex, with green upper wall composed of polyhedral cells, conidiogenous cells ampulliform, enteroblastic, and simple and curved conidia. All these features fit well with our fungus. *Lawalreea lecanorae* differs from our fungus in the larger conidiomata, 30–100 µm diam., larger conidiogenous cells and conidia, these last being (5–)5.5–6.5(–7.5) × 2–3 µm that are also curved. Furthermore, the host is different: apothecia and, rarely, thallus of *Lecanora persimilis*. For the moment, we think *Lawalreea* is the best genus for our fungus.

Etymology: We dedicate this species to Ana Rosa Burgaz on the occasion of her retirement, honoring her important lichenological work during her career and her kindness with her lichenological pupils and friends.

*Lawalreea burgaziana* is very abundant on pinkish zones of *Platismatia glauca* but very regularly distributed, covering large surfaces of the host. The pinkish color could be due to this species, but *Platismatia* has several other fungi in the treated area, like *Lichenoconium erodens*, *Lichenostigma maureri*, *Nesolechia oxyspora* or *Roselliniella* sp., so we cannot be sure what fungus is involved in the color change.

Loc. 10, J. Etayo 33,664 (hb. Etayo-topotype).

Sample of *Briancoppinsia cytospora* compared: SPAIN: Euzkadi, Álava, between Quintana and S. Román de Campezo, Izquiz forest, young trees of *Quercus pyrenaica*, 755 m, 42°40′10″ N, 2°27′22″ W, 26–Augut–2009, J.Etayo 25,311 (hb. Etayo, conf. P. Diederich).

***Licea parasitica*** (Zukal) G.W. Martin

This is an organism belonging to the Myxomycetes (Protista), which has been mentioned several times as lichenicolous. In Courel, it seems to be common for the locality presented below on *Evernia prunastri*.

Loc. 8, on *Evernia prunastri* growing on *Corylus avellana*, J. Etayo 33,364 (hb. Etayo).

***Lichenoconium cargillianum*** (Linds.) D. Hawksw.

This is a common species on apothecia of *Usnea florida*, which turn black and finally collapse. Conidiomata 100–160 µm diam. and conidia relatively large, subspherical, 5–6.5 µm diam., scarcely ornamented are characteristic. On this host, it was recorded by Hawksworth [34].

Loc. 1, on *Usnea florida* growing on *Castanea sativa*, J. Etayo 33,401 (hb. Etayo). Loc. 3a, on *Usnea* sp. growing on *Quercus petraea*, J. Etayo 33,417 (hb. Etayo).

***Lichenoconium erodens*** M.S. Christ. & D. Hawksw.

It is a very common species on *Evernia*, where it provokes pink- to violet-colored zones and eventually causes its death. *Platismatia glauca* is another host for this species in the studied area. Although there are several associated fungi that can colonize *Platismatia*, the most common is *Lichenoconium erodens* and maybe causes the pink color on many thalli, even on the galls produced by *Nesolechia oxyspora.* It is interesting that *Endococcus tricolorans* [38] fades the thallus of *Platismatia* in the same way, pink with an external black halo. On our samples, we found no trace of this species.

Loc. 1, on *Evernia prunastri* growing on *Castanea sativa*, hb. Etayo 33,389 (SANT-Lich. 12574). Ibidem, on *Usnea florida* growing on *C*, *sativa*, J. Etayo 33,403 (hb. Etayo). Loc. 2b, on *E. prunastri* growing on *Quercus* sp., J. Etayo 33,578 (SANT-Lich. 12580). Loc. 2b, on *Pseudevernia furfuracea* growing on *Corylus avellana*, J. Etayo 33,586 (SANT-Lich. 12582). Ibidem, on *P. furfuracea*, J. Etayo 33,590 (hb. Etayo). Loc. 2b, on *Usnea* sp. growing on *C. avellana*, J. Etayo 33,587 (SANT-Lich. 12579). Loc. 3a, on *E. prunastri* growing on *Quercus petraea*, J. Etayo 33,438 (SANT-Lich. 12594). Ibidem, on *Alectoria sarmentosa* growing on *Quercus petraea*, J. Etayo 33,446 (hb. Etayo). Ibidem, on *Platismatia glauca* growing on *Q. petraea*, J. Etayo 33,458 (hb. Etayo). Loc. 7, on *E. prunastri* growing on *Betula pubescens*, J. Etayo 33,468 (hb. Etayo). Loc. 8, on *E. prunastri* growing on *C. avellana*, J. Etayo 33364, 33,365 (hb. Etayo). Loc. 9, on *Parmelia sulcata* and *P. glauca* growing on *Fagus sylvatica*, J. Etayo 33,645 (SANT-Lich. 12612). Ibidem, on *P. glauca* on *B. pubescens*, M.E. López de Silanes (SANT-Lich. 12538). Ibidem, on *P. sulcata*, growing on twigs of *Taxus baccata*, M.E. López de Silanes (SANT-Lich. 12704). Ibidem on *P. sulcata*, growing on *Quercus pyrenaica*, M.E. López de Silanes (SANT-Lich. 12706). Loc. 10, on *P. glauca* growing on *F. sylvatica*, J. Etayo 33,649 (hb. Etayo, SANT-Lich. 12618). Ibidem on *P. glauca* growing on *I. aquifolium*, M.E. López de Silanes (SANT-Lich. 12545). Loc. 11, on *Flavoparmelia caperata* growing on *C. sativa*, J. Etayo 33,485 (hb. Etayo). Ibidem, on *Lecanora intumescens* growing on *C. sativa*, J. Etayo 33,486 (SANT-Lich. 12602). Loc. 13, on *Ramalina fastigiata* growing on *C. sativa*, J. Etayo 33,544 (hb. Etayo). Ibidem, on *E. prunastri*, J. Etayo 33,546 (SANT-Lich. 12624). Ibidem, on *F. caperata* growing on *C. sativa*, J. Eayo 33,550 (hb. Etayo).

***Lichenoconium lecanorae*** (Jaap) D. Hawksw.

Loc. 9, on apothecia of *Parmelia saxatilis* growing on *Fagus sylvatica*, J. Etayo 33,633 (SANT-Lich. 12635).

***Lichenoconium pyxidatae*** (Oudem.) Petr. et Syd.

Loc. 10, on *Cladonia pyxidata* growing on limestone rocks, J. Etayo 33,662 (hb. Etayo).

***Lichenoconium usneae*** (Anzi) D. Hawksw.

We distinguish this species from *L. cargillianum*, which we also found on *Usnea* sp., on the basis of its smaller conidiomata and conidia, these 3–4 µm diam.

Loc. 3a, on *Evernia prunastri* growing on *Quercus petraea*, J. Etayo 33,422 (SANT-Lich. 12598). Ibidem, on *Usnea florida* growing on *Q. petraea*, J. Etayo 33,448 (hb. Etayo). Loc. 7, on *Xanthoria parietina* growing on *Sambucus nigra*, J. Etayo 33,471 (hb. Etayo).

***Lichenodiplis lecanorae*** (Vouaux) D. Hawksw.

Loc. 7, on *Athallia cerinelloides* growing on *Sambucus nigra*, J. Etayo 33,472 (hb. Etayo).

***Lichenodiplis pertusariicola*** (Nyl.) Diederich

Loc. 12, on *Pertusaria leioplaca* growing on *Castanea sativa*, J. Etayo 33,562 (hb. Etayo).

***Lichenopeltella ramalinae*** Etayo & Diederich

It seems to be a very common species, usually on *Ramalina farinacea* in high and humid forests, particularly on beeches in Courel. Thalli of *Ramalina* are blackened due to the infection with this fungus. Sometimes, it is hyperparasitized by *Spirographa giselae*. On trunks where we found it together *R. farinacea* and other species like *R. fastigiata*, it appears only on the first one.

Loc. 1, on *Ramalina farinacea* growing on *Castanea sativa*, hb. Etayo 33,397 (hb. Etayo). Loc. 2b, on *R. farinacea* growing on *Corylus avellana*, J. Etayo 33,593 (SANT-Lich. 12581). Loc. 3a, on *R. farinacea* growing on *Quercus petraea*, J. Etayo 33,419 (SANT-Lich. 12595). Ibidem, on *R. farinacea* growing on *Q. petraea*, M.E. López de Silanes (SANT-Lich. 12493). Loc. 8, on *R. farinacea* growing on *Fagus sylvatica*, *Laurus nobilis* and *C. avellana*, J. Etayo 33,354 (hb. Etayo). Loc. 9, on *R. farinacea* growing on *F. sylvatica*, J. Etayo 33,635 (SANT-Lich. 12611, hb. Etayo). Loc. 11, on *R. farinacea* growing on *C. sativa*, J. Etayo 33,484 (hb. Etayo). Loc. 12, on *R. farinacea* growing on *C. sativa*, J. Etayo 33,574 (SANT-Lich. 12605). Loc. 13, on *R. farinacea* growing on *C. sativa*, J. Etayo 33,544 (hb. Etayo). Ibidem, on *R. farinacea* growing on *Q. petraea*, M.E. López de Silanes (SANT-Lich. 12493).

***Lichenopuccinia poeltii*** D. Hawkw. & Hafellner

It is a very rarely recorded species in the Iberian Peninsula, but it seems to be quite common in some locations in Lugo province.

Loc. 2b, on *Parmelia saxatilis* growing on *Corylus avellana*, J. Etayo 33,588 (hb. Etayo). Loc 3a, on *P. saxatilis* growing on *Quercus petraea*, M.E. López de Silanes (SANT-Lich. 12699). Loc. 9, on *P. saxatilis* growing on *Fagus sylvatica*, J. Etayo 33,616 (hb. Etayo). Ibidem, on *Parmelia sulcata* growing on *F. sylvatica*, J. Etayo 33,631 (hb. Etayo).

***Lichenostigma*** aff. ***alpinum*** (R. Sant., Alstrup & D. Hawksw.) Ertz & Diederich

Etayo [39] described *Diederimyces fuscideae* growing on corticolous thalli of *Fuscidea cyathoides* in oceanic zones of northern Spain and southern France, as the sexual morph of *Phaeosporobolus alpinus*. The name, however, was invalid. Ertz et al. [40] considered the possibility that it was a synonym of *Lichenostigma alpinum*.

Loc. 3b, on *Fuscidea cyathoides* growing on *Betula pubescens*, J. Etayo 33,437 (hb. Etayo).

***Lichenostigma chlaroterae*** (Berger & Brackel) Ertz & Diederich

Loc. 0, on *Lecanora chlarotera* growing on twig of *Acer* sp., J. Etayo 33,460 (SANT-Lich. 12626). Loc. 3a, on *Lecanora intumescens* growing on *Quercus petraea*, J. Etayo 33,455 (hb. Etayo). Loc. 7, on *L. chlarotera* growing on *Prunus cerasus*, J. Etayo 33,473 (hb. Etayo).

***Lichenostigma maureri*** Hafellner

Typically growing on *Usnea*, we also found it on *Evernia prunastri*, *Hypogymnia tubulosa*, *Platismatia glauca* and *Pseudevernia furfuracea*.

Loc. 0, on *Hypogymnia tubulosa* growing on twig of *Acer* sp., J. Etayo 33,462 (hb. Etayo). Loc.1, on *Usnea florida* growing on *Castanea sativa*, J. Etayo 33,382 (hb. Etayo). Ibidem, on *Pseudevernia furfuracea* growing on *C. sativa*, hb. Etayo 33,393 (SANT-Lich. 12571). Ibidem, on *U. florida* growing on *C. sativa*, J. Etayo 33,402 (hb. Etayo). Loc. 2b, on *Usnea* sp. growing on *Quercus* sp., J. Etayo 33,576 (hb. Etayo). Loc. 2b, on *P. furfuracea* growing on *Corylus avellana*, J. Etayo 33,586 (SANT-Lich. 12582). Ibidem, on *P. furfuracea* growing on *C. avellana*, J. Etayo 33,590 (hb. Etayo). Loc. 3a, asexual morph on *Evernia prunastri* growing on *Quercus petraea*, J. Etayo 33,422 (SANT-Lich. 12598). Ibidem, on *P. furfuracea* growing on *Q. petraea* and *Betula pubescens*, J. Etayo 33,425 (hb. Etayo). Ibidem, on *U. florida* growing on *Q. petraea*, J. Etayo 33,448 (hb. Etayo). Ibidem, on *Platismatia glauca* growing on *Q. petraea*, J. Etayo 33,459 (hb. Etayo). Loc. 9, on *Hypogymnia tubulosa* growing on *Quercus pyrenaica*, J. Etayo 33,617 (hb. Etayo). Ibidem, on *P. furfuracea* growing on *Q. pyrenaica*, J. Etayo 33,618 (hb. Etayo). Ibidem, on *E. prunastri* growing on *Fagus sylvatica*, J. Etayo 33,629 (hb. Etayo).

***Lichenostigma*** sp.

Two sterile, probably undescribed, species of this genus were found growing on saxicolous lichens.

Loc. 4, on *Rhizocarpon geographicum* growing on exposed rocks, J. Etayo 33,598 (hb. Etayo, SANT-Lich. 12567). Loc. 4, on *Porpidia* sp. Growing on rocks, J. Etayo 33,611 (hb. Etayo).

***Micarea* *amplissima*** van den Boom & Etayo

A recently described species that is present in France [9] and Spain (Álava, Asturias, León, Lugo, and Navarra). It always grows on thallus and apothecia of *Ricasolia amplissima.* Its dispersed greenish, often darkened to black goniocysts, c. 25–30 µm wide, are characteristic.

Loc. 1, on *Ricasolia amplissima* growing on *Castanea sativa*, J. Etayo 33,375 (hb. Etayo). Loc. 3a, on *R. amplissima* growing on *Quercus petraea*, M.E. López de Silanes (SANT-Lich. 12457, 12700). Loc. 9, on *R. amplissima* growing on *Fagus sylvatica*, J. Etayo 33,626 (hb. Etayo). Ibidem, M.E. López de Silanes (SANT-Lich. 12459). Loc. 10, on *R. amplissima* growing on undetermined trunk, J. Etayo 33668.

***Micarea* *peliocarpa*** (Anzi) Coppins & R. Sant.

This is a mainly corticolous species [9], recorded as lichenicolous on *Hypotrachyna*, *Pertusaria*, and *Ricasolia* from France, Portugal, and Spain (Gipuzkoa, Lugo, and Salamanca). In Lugo, we found it on *Melanelixia*, *Parmelia*, and *Phlyctis* as well.

Loc. 8, on *Parmelia saxatilis* growing on *Fagus sylvatica*, J. Etayo 33,352 (hb. Etayo). Ibidem, on *Phlyctis argena* growing on *Corylus avellana*, J. Etayo 33,359 (hb. Etayo). Ibidem, on *Melanelixia glabratula* growing on twigs of *F. sylvatica*, J. Etayo 33,357 (hb. Etayo). Loc. 9, on *Ricasolia amplissima* growing on *Quercus robur*, J. Etayo 33,623 (SANT-Lich. 12457, hb. Etayo).

***Micarea usneae*** van den Boom & Ertz

This species was described from Madeira [41] and later recorded in Galicia (Spain) [9], accounting for the first report from mainland Europe.

Loc. 9, on *Usnea florida* growing on *Fagus sylvatica*, J. Etayo 33,643 (hb. Etayo).

***Microsphaeropsis olivacea*** (Bonord.) Höhn.

The size and shape of conidia of our sample living on *Parmelia sulcata* fit well with the previous description on other hosts such as *Everniastrum*, *Everniopsis*, *Hypotrachyna*, *Parmelia*, *Parmotrema*, *Sticta*, and *Yoshimuriella* [42]. On *Parmelia sulcata* it causes an infection surrounded by a black line, where conidiomata are firstly immersed, then erumpent, breaking the host cortex.

Loc. 3b, on *P. sulcata* and *P. saxatilis* growing on *Betula pendula*, J. Etayo 33,426 (hb. Etayo).

***Milospium lacoizquetae*** Etayo & Diederich

We found this species growing alongside *Arthonia digitata* on *Cladonia digitata*, which is a common occurrence for this species [10,43].

Loc. 1, on squamules and podetia of *C. digitata* growing on wood of *Castanea sativa*, J. Etayo 33,384 (hb. Etayo). Loc. 3a, on *C. digitata* growing on *Quercus petraea*, J. Etayo 33,429 (SANT-Lich. 12596). Loc. 11, on *C. digitata* growing on wood of *C. sativa*, J. Etayo 33,478 (hb. Etayo).

***Minutoexcipula tephromelae*** V. Atienza, Etayo & Pérez-Ortega

This species was described in Atienza et al. [44] on saxicolous *Tephromela atra* from La Rioja and Soria in Spain. This is the first Spanish record on corticolous *Tephromela atra*.

Loc. 8, on *Tephromela atra* growing on *Fraxinus* sp., J. Etayo 33,359 (hb. Etayo).

***Monerolechia badia*** (Fr.) Kalb

Loc. 6, on *Xanthoparmelia conspersa* growing on acid slope, J. Etayo 33,495 (hb. Etayo).

***Muellerella erratica*** (A. Massal.) Hafellner & V. John

Loc. 4, on *Lecanora polytropa* growing on exposed rocks, J. Etayo 33,600 (hb. Etayo). Ibidem, on *Lecanora sulphurea* growing on a boulder, J. Etayo 33,610 (hb. Etayo).

***Muellerella ventosicola*** (Mudd) D. Hawksw.

Loc. 4, very common between *Rhizocarpon geographicum* thalline areoles growing on exposed rocks, J. Etayo 33,598 (hb. Etayo, SANT-Lich. 12567).

***Myxotrichum bicolor*** (Ehrenb. Ex Pers.) Fr.

This species was recorded in Clauzade et al. [45] as possibly only fortuitously lichenicolous on *Cetraria* and *Usnea*. On this last host, we found it on several occasions (but not published) in Navarra, but this is the first time we collected it on *Evernia prunastri*.

Loc. 8, on *Evernia prunastri* growing on *Corylus avellana*, J. Etayo 33,366 (hb. Etayo).

***Nanostictis christiansenii*** Etayo

Described in Etayo and Diederich [43], this species seems to be uncommon in the studied localities as its presence was very scarce among the considerable number of unhealthy *Lobaria pulmonaria* thalli that were examined.

Loc. 9, on *Lobaria pulmonaria* growing on *Fagus sylvatica*, J. Etayo 33,625 (hb. Etayo).

***Nectriopsis lecanodes*** (Ces. in Rabenh.) Diederich

This species is very abundant, especially on *Castanea* woods, where it lives on thalli of *Lobaria*, *Lobarina*, *Nephroma*, etc. On *Ricasolia amplissima*, it causes a remarkable zonation of the host’s thalli and can grow even on its cyanobacterial morph (*Dendriscocaulon*).

Loc. 1, on *Lobaria pulmonaria* growing on *Castanea sativa*, J. Etayo 33,373 (hb. Etayo). Ibidem, on *Lobarina scrobiculata*, J. Etayo 33,374 (hb. Etayo). Ibidem, on *L. scrobiculata* growing on *C. sativa*, M.E. López de Silanes (SANT-Lich 12458). Ibidem, on *Nephroma laevigatum* growing on *C. sativa*, J. Etayo 33,377 (hb. Etayo). Ibidem, on *Ricasolia amplissima* growing on *C. sativa*, J. Etayo 33,378 (hb. Etayo). Ibidem, on *Peltigera membranacea* growing on *C. sativa*, J. Etayo 33,390 (hb. Etayo). Loc. 9, on *L. pulmonaria* growing on *Fagus sylvatica*, J. Etayo 33,641 (hb. Etayo). Loc. 10, on *N. laevigatum* growing on *Prunus* sp., J. Etayo 33,654 (hb. Etayo). Ibidem, on *L. pulmonaria* growing on *F. sylvatica*, J. Etayo 33,656 (SANT-Lich. 12616). Loc. 11, on *L. scrobiculata* and *N. laevigatum* growing on *C. sativa*, J. Etayo 33,476 (hb. Etayo). Ibidem, on *L. scrobiculata* growing on *C. sativa*, J. Etayo 33,479 (hb. Etayo). Loc. 12, on unhealthy *R. amplissima* growing on *C. sativa*, J. Etayo 33,566 (SANT-Lich. 12606). Ibidem on *R. amplissima* on *C. sativa*, M.E. López de Silanes (SANT-Lich 12558).

***Nesolechia fusca*** (Triebel & Rambold) Pérez-Ortega

Loc. 2b, on *Hypogymnia tubulosa* growing on branches, J. Etayo 33,583 (hb. Etayo). Loc. 3b, on *H. tubulosa* growing on twigs of *Betula pubescens*, J. Etayo 33,436 (hb. Etayo). Loc. 9, on *H. tubulosa* growing on *Quercus pyrenaica*, J. Etayo 33,617 (hb. Etayo). Ibidem on *H. tubulosa* growing on twigs of *Q. pyrenaica*, M.E. López de Silanes (SANT-Lich 12707)

***Nesolechia oxyspora*** (Tul.) A. Massal. var. oxyspora

It is a very common species, forming large and abundant, sometimes cerebriform, sometimes concolorous with thallus, pink- or mauve-colored galls on large and with abundant apothecia *Platismatia glauca* and more rarely on *Parmelia sulcata.*

Loc. 3a, on *Platismatia glauca* growing on *Quercus petraea*, M.E. López de Silanes (SANT-Lich 12492). Loc. 9, on *Parmelia sulcata* growing on *Fagus sylvatica*, J. Etayo 33,631 (hb. Etayo). Ibidem on *P. glauca* growing on *Betula pubescens*, M.E. López de Silanes (SANT-Lich 12538). Loc. 10, very common on *P. glauca* growing on *F. sylvatica*, J. Etayo 33,649 (hb. Etayo, SANT-Lich 12618).

***Niesslia lobariae*** Etayo & Diederich

Described in Etayo and Diederich [43], we found it relatively commonly in the study area.

Loc. 9, on *Lobaria pulmonaria* growing on *Fagus sylvatica*, J. Etayo 33,622 (SANT-Lich. 12613). Ibidem, J. Etayo 33,642 (hb. Etayo). Loc. 10, on *L. pulmonaria* growing on *F. sylvatica*, J. Etayo 33,656 (SANT-Lich. 12616). Loc. 12, on unhealthy *L. pulmonaria* growing on *Castanea sativa*, J. Etayo 33,558 (hb. Etayo).

***Normandina pulchella*** (Borrer) Nyl.

This is a lichen that, partially due to its small squamules, can easily cover other lichens.

Loc 9, on *Ricasolia amplissima* on *Fagus sylvatica*, M.E. López de Silanes (SANT-Lich. 12700). Loc. 11, on *Lobarina scrobiculata* on *Castanea sativa*, J. Etayo 33,476 (hb. Etayo).

***Opegrapha anomea*** Nyl.

It seems to be common in the studied area, growing on *Lepra*.

Loc. 1, on *Lepra albescens* growing on *Castanea sativa*, J. Etayo 33,371 (hb. Etayo). Loc. 3a, on *L. albescens* growing on *Quercus petraea*, J. Etayo 33,432 (SANT-Lich. 12597). Loc. 9, on *L. albescens* growing on *Fagus sylvatica*, J. Etayo 33,644 (hb. Etayo).

***Opegrapha sphaerophoricola*** Isbrad & Alstrup

This species was abundant on a vertical rock in a forest covered by *Sphaerophorus globosus*. We also found the host on trees with no presence of the fungus. In both cases, there were some common pink zones on the host.

Loc. 3a, on *S. globosus* growing on vertical rock, J. Etayo 33,427 (hb. Etayo). Ibidem, M.E. López de Silanes (SANT-Lich 12495).

***Paranectria oropensis*** (Cesati) D. Hawksw.

Loc. 10, very rare on *Pannaria conoplea* and *Lopadium disciforme* growing on a tree, J. Etayo 33,665 (hb. Etayo).

***Plectocarpon lichenum*** (Sommerf.) D. Hawksw.

Loc. 9, on *Lobaria pulmonaria* growing on *Fagus sylvatica*, J. Etayo 33,621 (hb. Etayo, SANT-Lich. 12614). Ibidem, on *L. pulmonaria* growing on *Fagus sylvatica*, J. Etayo 33,627 (hb. Etayo). Ibidem on *L. pulmonaria* on *F. sylvatica*, M.E. López de Silanes (SANT-Lich 12536).

***Plectocarpon scrobiculatae*** Diederich & Etayo

It appears to be uncommon on the Iberian Peninsula. It was previously observed in Ancares [7].

Loc. 3a, on *Lobarina scrobiculata* growing on *Quercus petraea*, J. Etayo 33,413 (hb. Etayo).

***Pronectria anisospora*** (Lowen) Lowen

≡ *Trichonectria anisospora* (Lowen) Diederich & van den Boom

Loc. 2b, on *Hypogynmia tubulosa* growing on branches of undetermined tree, J. Etayo 33,583 (hb. Etayo). Loc. 3a, on *H. tubulosa* growing on *Quercus petraea*, J. Etayo 33,450 (hb. Etayo). Loc. 9, on *H. tubulosa* growing on *Quercus pyrenaica*, J. Etayo 33,617 (hb. Etayo). Loc. 11, on *H. physodes* growing on *Castanea sativa*, J. Etayo 33,482 (hb. Etayo).

***Pronectria fragmospora*** Etayo

Up to now, this species was exclusively known from Navarino and Punta Arenas, Chile [16]. The Galician sample presented below, growing on a pendulous, sorediate *Usnea*, is almost identical to the description. Ascomata are immersed, rarely erumpent, orange pinkish, 100–150 µm diam. Hymenium inspersed with vivid orange-colored droplets. Asci 8-spored. Ascospores breaking into semispores within the ascus, 10–16 × 5.5–7.5 µm, a bit broader than described (10–15 × 5–6 µm). The first author has other samples from Spain: Huesca and León (unpublished).

Loc. 2b, on *Usnea* sp. growing on *Corylus avellana*, J. Etayo 33,590 (hb. Etayo).

***Pronectria pertusariicola*** Lowen

Loc. 1, on *Lepra albescens* growing on *Castanea sativa*, hb. Etayo 33,395 (SANT-Lich. 12572). Loc. 10, on *L. albescens* growing on mossy trunk, J. Etayo 33,666 (hb. Etayo).

***Pronectria scrobiculatae*** Etayo & López de Silanes, sp. nov. (Figure 3)

MycoBank MB 851492

Differs from the similar *P. robergei* in its smaller perithecia, 180–200 µm vs. 240–340 µm diam., with smooth and more widely ellipsoid spores (10–)11–12.5(–14) × (6–)6.5–7.5(–9) μm vs. ascospores finally verruculose, elongated (very variable) 10.5–20 × 2.5–7.5 μm. Additionally, *Pronectria robergei* only grows on *Peltigera* and develops a specific anamorphic stage *Illosporium carneum* never found on *Lobarina scrobiculata* the host of *Pronectria scrobiculatae*.

TYPUS. SPAIN, Lugo, Courel, village, from Seoane to Paderne by the Pequeno River, on *Lobarina scrobiculata* growing on *Castanea sativa*, 42°38′58.9″ N 7°09′35.1″ W, 620–650 m, 6–July–2021, J. Etayo 33479 and M.E. López de Silanes (SANT–Lich. 12715–holotype designated here, hb. Etayo–isotype).

Perithecia obpyriform to subglobose, with a small and concolorous papilla on the top, protruding out of the host thallus, orange brown to dark brown, 180–200 µm diam. Ascomatal wall orange to brownish olivaceous, 30–40 µm thick, composed of several layers, hyaline inside and olive orangish outside, K-, KI- or intensifying the orangish color. The outer layer composed by subglobose and large cells, 10–20 µm diam. Papilla yellow in section, formed by a dense group of concolorous hyphae with thickened wall, 2–3 µm thick. Ostiolar canal with numerous periphyses, 1–1.5 µm wide. Hymenium hyaline, KI-, with interascal filaments with very thin wall, composed of large ellipsoid cells, 3–11 µm wide, filled with many small oil droplets. Asci cylindrical, with subtruncate end and a thickened apical zone, 8-spored, 75–95 × 8–10 µm (n = 7). Ascospores uniseriate in the ascus, broadly ellipsoid to slightly rhomboidal, (0–)1–septate, not or hardly constricted at the septum, hyaline even when overmature, smooth-walled, colorless, with one large oil droplet in each cell, (10–)11–12.5(–14) × (6–)6.5–7.5(–9) µm (n = 30).

This species is clearly pathogenic as the infected lichen thallus becomes papyraceous and grey greenish. It is only known from the type locality.

No other species of *Pronectria* or *Xenonectriella* were recorded growing on the genus *Lobarina*, so there is no doubt it is a new species. However, we placed in the genus *Pronectria* with some hesitation due to some coincident features with *Xenonectriella*, especially the ascus type, large, thickened in upper zone and with uniseriate, widely ellipsoid spores. However, the two main features of this genus that separate it from *Pronectria* are a dark-colored wall, K+ intense purple or violet and ornamented, finally brownish to golden-brown ascospores. None of these are present in *P. scrobiculatae*. Ascospores in *P. scrobiculatae* look like young ascospores of *Xenonectriella*, which are later ornamented and darken. However, we have observed many of them, and they are always colorless even in overmature spores.

The closest species of *Pronectria* to the described species are *P. robergei* and *P. loweniae*. In its compilation, Brackel [12] considered *P. robergei* as inhabiting the species of *Peltigera* and *Solorina*. However, its specimens growing on *Solorina* were recently described as *P. loweniae* [46]. Compared to *Pronectria scrobiculatae*, *P. robergei* has larger perithecia, 240–340 µm diam., with more elongated (very variable) ascospores, 10.5–20 × 2.5–7.5 μm [47] finally verruculose. Additionally, *Pronectria robergei* develops specific anamorphic stage *Illosporium carneum* [48,49], occurring on various species of *Peltigera*, but never reported on other genera of Peltigerales [12]. *Pronectria loweniae* differs from *P. scrobiculatae* in its ascomata being yellowish orange, with projecting hyphae radially spreading; ascomatal wall cells strongly inspersed by small orange oil droplets; smaller asci 60–70 × 8–10 µm and ascospores generally more elongated, also variable in size 10–18 × 3–8 μm

Etymology: The epithet “*scrobiculatae*” refers to the host lichen *Lobarina scrobiculata*, a common species in the studied forests.

***Raesaenenia huuskonenii*** (Räsänen) D. Hawksw., C. Boluda & H. Lindgr.

We report a sample of *Bryoria* sp. with galls morphologically similar to the sexual morph of this species but only with grouped conidiomata forming a stroma, with upper part greenish and the lower brown, with the same pigments as those in the ascomata of this species. Conidiogenous cells ampulliform to cylindrical, conidia slightly sinuous and fusiform, 6–9 × 2 µm.

Loc. 9, on *Bryoria* sp. growing on *Fagus sylvatica*, J. Etayo 33,638 (hb. Etayo).

***Ramboldia insidiosa*** (Th. Fr.) Hafellner

Loc. 5, on *Lecanora varia*, on wooden fence, M.E. López de Silanes (SANT-Lich 12520).

***Refractohilum galligenum*** D. Hawksw.

We found it forming its characteristic convex galls on the thallus of *Nephroma*. Younger galls usually have conidiophores and conidia. Sometimes, we found it in the galls also small conidiomata with thin conidia of c. 7–8 × 1.5 µm belonging to another undetermined species.

Loc. 10, on *Nephroma laevigatum* growing on *Prunus*, J. Etayo 33,654 (hb. Etayo). Ibidem on *N. laevigatum* growing on *Corylus avellana*. M.E. López de Silanes (SANT-Lich 12543). Loc. 11, on *N. laevigatum* growing on *Castanea sativa*, J. Etayo 33,476 (hb. Etayo).

***Refractohilum intermedium*** Roux & Etayo

We found this cryptic species on thallus and apothecia of *Gyalecta carneola* (Ach.) Hellb.

Loc. 3a, on *G. carneola* growing on *Quercus petraea*, J. Etayo 33,442 (hb. Etayo).

***Rhymbocarpus aggregatus*** Etayo & Diederich

This species was described from the Iberian Peninsula growing on *Buellia griseovirens* [50]. It is characterized by its grouped apothecia, black, sessile, roundish, 100–160 µm diam., with a dark orange to chestnut brown, K+ (slightly darker brown) exciple; epihymenium olivaceous greenish, K+ intensifying and hymenium greenish to colorless, 35–45 µm tall; asci elongate-clavate to subcylindrical, apically rounded, 8-spored, 31–42 × 6–8 µm and ascospores simple, straight, 7–11.5 × 2.2–3.2 µm (in the Galician sample). It was previously known only from Navarra and Aragón in Spain [51].

Loc. 3a, on *Buellia griseovirens* growing on *Quercus petraea*, J. Etayo 33,455 (hb. Etayo).

***Rhymbocarpus neglectus*** (Vain.) Diederich & Etayo

According to Diederich and Etayo [52], it is a parasitic fungus on *Lepraria* gr. *neglecta*, and it is known from several European countries, arctic Russia, and Mongolia [53]. The nearest locality is in Serra da Estrela, Portugal [52].

Loc. 2a, on *Lepraria* sp. on twig of Ericacea, J. Etayo 33,670 (hb. Etayo).

***Rinodina efflorescens*** Malme

Corticolous lichen occasionally overgrowing other lichens, especially *Parmelia sulcata* in the region. It prefers growing on thallus border where its small, sorediate squamules can be seen. In the Iberian Peninsula, it has been noted growing on other lichens [54].

Loc. 2b, on *P. sulcata* growing on *Corylus avellana*, J. Etayo 33,592 (hb. Etayo). Loc. 3a, on *Parmelia saxatilis* and *P. sulcata* growing on *Quercus petraea*, J. Etayo 33,430 (hb. Etayo). Ibidem, on *Melanelixia glabratula* growing on *Q. petraea*, J. Etayo 33,451 (SANT-Lich. 12592).

***Roselliniella*** cf. ***atlantica*** Matzer & Hafellner

We found similar perithecia to those of this species growing isolated or in small groups on *Parmelia saxatilis* and *Platismatia glauca*. Perithecia are firstly immersed, then erumpent and finally sometimes sessile, in the latter case with abundant visible setae. Described in Matzer and Hafellner [55] as parasitic on *Xanthoparmelia mougeotii* and *X. verruculifera* from Europe.

Loc. 2b, on *Parmelia saxatilis* growing on *Quercus* sp., J. Etayo 33,575 (hb. Etayo). Loc. 8, on *P. saxatilis* growing on *Fagus sylvatica*, J. Etayo 33,355 (hb. Etayo). Loc. 10, on *P. saxatilis* growing on *F. sylvatica*, J. Etayo 33,658 (hb. Etayo). Loc. 8, on *Platismatia glauca* growing on *Corylus avellana*, J. Etayo 33,368 (hb. Etayo).

***Sclerococcum lobariellum*** (Nyl.) Ertz & Diederich

Loc. 9, on *Lobaria pulmonaria*, on *Fagus sylvatica*, M.E. López de Silanes (SANT-Lich 12540). Loc. 12, on *L. pulmonaria* on *Castanea sativa*, J. Etayo 33,558 (hb. Etayo).

***Sclerococcum montagnei*** Hafellner

Loc. 4, on *Lecanora rupicola* growing on exposed boulder, J. Etayo 33,605 (hb. Etayo).

***Sclerococcum sphaerale*** Fr.

Loc. 4, on *Pertusaria corallina* growing on exposed boulder, J. Etayo 33,606 (hb. Etayo).

***Sphaerellothecium parmeliae*** Diederich & Etayo

We commonly found it in the study area living on two corticolous species of *Parmelia*, *P. saxatilis*, *P. sulcata* and more rarely on the saxicolous species *P. omphalodes*.

Loc. 3a, on *P. saxatilis* and *P. sulcata* growing on *Quercus petraea*, J. Etayo 33,430 (hb. Etayo). Ibidem on *P. sulcata* growing on *Q. petraea*, M.E. López de Silanes (SANT-Lich. 12698). Loc. 3b, on *P. sulcata* growing on *Betula pubescens*, M.E. López de Silanes (SANT-Lich 12500). Loc. 4, on *Parmelia omphalodes* growing on exposed boulder, J. Etayo 33,603 (hb. Etayo).

***Sphinctrina anglica*** Nyl.

We found it on the thallus of *Protoparmelia ochrococca*, the same host species as in the finds from the British Isles [15].

Loc. 12, on *P. ochrococca* growing on wood of *Castanea*, J. Etayo 33,570 (hb. Etayo).

***Sphinctrina leucopoda*** Nyl.

Loc. 1, on *Pertusaria* cf. *leioplaca* growing on *Ilex aquifolium*, J. Etayo 33,396 (SANT-Lich. 12573).

***Spirographa giselae*** (Brackel) Flakus, Etayo & Miadlikowska

This hyperparasitic species lives on black thalli and ascomata of *Lichenopeltella ramalinae*. It was known from Bolivia, Portugal, and Spain; in Spain, it was recorded from Cáceres and Madrid provinces [56].

Loc. 1, on *Lichenopeltella ramalinae* settled on *Ramalina farinacea* growing on *Castanea sativa*, hb. Etayo 33,397 (hb. Etayo). Loc. 11, on *L. ramalinae* parasite on *R. farinacea* growing on *C. sativa* J. Etayo 33,484 (hb. Etayo).

***Spirographa*** spp.

We found two species of *Spirographa* not yet described in the studied region. The first one was growing on apothecia of *Melanohalea exasperata*, and it developed dispersed, subglobose, erumpent, black pycnidia, with Y-morph conidia similar to those of *S. usneae* [56]. The second one was growing on *Lecanora chlarotera* apothecia. Apothecia were small and ascospores curved, 1-septate, 17–25 × 2.5–3 µm.

Loc. 2b, on *Melanohalea exasperata* growing on *Quercus*, J. Etayo 33,579 (hb. Etayo). Loc. 7, on apothecia of *Lecanora chlarotera* growing on old *Prunus cerasus*, J. Etayo 33,473 (hb. Etayo).

***Stigmidium degelii*** R. Sant.

Loc. 3a, on *Pectenia plumbea* growing on *Quercus petraea*, J. Etayo 33,420 (hb. Etayo).

***Stigmidium peltideae*** (Vain.) R. Sant.

Loc. 9, on *Peltigera membranacea* growing on *Fagus sylvatica*, J. Etayo 33,619 (hb. Etayo). Loc. 10, on *P. neckeri* growing on soil, J. Etayo 33,657 (hb. Etayo).

***Stigmidium psorae*** (Anzi) Hafellner

Loc. 10, on *Psora decipiens* growing on exposed limestone boulder, J. Etayo 33,651 (hb. Etayo).

***Taeniolella delicata*** M.S. Christ. & D. Hawksw.

This is a common species growing on a large spectrum of not related lichens [57]. The identification of the specimen on *Buellia griseovirens* is questionable.

Loc. 2b, on *Physcia aipolia* growing on *Quercus*, J. Etayo 33,581 (hb. Etayo). Loc. 7, on *Buellia griseovirens* growing on *Prunus cerasus*, J. Etayo 33,473 (hb. Etayo).

***Toninia plumbina*** (Anzi) Hafellner & Timdal

This rare species seems to be frequent on thalli of *Pectenia plumbea* in the recorded locality. It was already recorded in Ancares by Álvarez et al. [7].

Loc. 3a, on *Pectenia plumbea* growing on *Quercus petraea*, J. Etayo 33,421 (hb. Etayo, SANT-Lich 12494).

***Trapeliopsis flexuosa*** (Fr.) Coppins & P. James

Lichenized fungus that we found growing on *Parmelia* in one locality.

Loc. 8, on *P. saxatilis* growing on *Fagus sylvatica*, J. Etayo 33,352 (hb. Etayo).

***Tremella cetrariicola*** Diederich & Coppins

It is quite common in Ancares growing on *Nephromopsis chlorophylla* (the same host species as in the type specimen of this parasite). It is known from many places in the Northern Hemisphere, including Canary Islands, but it was not recorded from mainland Spain [58].

Loc. 3a, on *Nephromopsis chlorophylla* growing on *Quercus petraea*, J. Etayo 33,409 (hb. Etayo, SANT-Lich. 12484).

***Tremella hypogymniae*** Diederich

This species is really common in Caurel and Ancares, especially on unhealthy thalli of *Hypogymnia tubulosa*. See distribution of this species in [58].

Loc. 3b, on *Hypogymnia tubulosa* growing on *Betula pubescens*, J. Etayo 33,428 (hb. Etayo). Loc. 8, on *H. tubulosa* growing on *Corylus avellana* and branches of *Fagus sylvatica*, J. Etayo 33,363 (hb. Etayo). Loc. 9, on *H. tubulosa* growing on *Quercus pyrenaica*, J. Etayo 33,617 (hb. Etayo, SANT-Lich. 12705). Loc. 11, on *H.* cf. *tubulosa* growing on *Castanea sativa*, J. Etayo 33,482 (hb. Etayo).

***Tremella*** sp.

We found some thalli of *Melanohalea exasperata* with small galls. Inside there are not basidia, but structures that look like conidia. We have another specimen of this species from Moncayo (Zaragoza).

Loc. 2b, on *Melanohalea exasperata* growing on *Corylus avellana*, J. Etayo 33,597 (hb. Etayo).

***Trichonectria*** cf. ***australis*** Etayo

We found two specimens probably belonging to this species as described by Etayo and Sancho [16]. They have small, grouped, firstly whitish then brown, cupuliform ascomata 100–120 µm diam.; hairs distributed around the ostiole; ascospores 1-septate, ellipsoid to cylindrical, with obtuse ends, easily broken at the septum, 10–13 × 3–3.5 µm. As the diversity of *Trichonectria* growing on *Usnea* is really high, we provisionally refer the examined material to *T.* cf. *australis.*

Loc. 8, on *Usnea* gr. *florida* growing on *Fagus sylvatica*, J. Etayo 33,353 (hb. Etayo). Loc. 9, on *Usnea* sp. growing on *F. sylvatica*, J. Etayo 33,639 (hb. Etayo).

***Trichonectria parmeliellae*** Etayo, sp. nov. (Figure 4)

MycoBank MB 851493

Differs from all other lichenicolous species of the genus *Trichonectria* in its orange, globose, hairy perithecia 130–180 µm diam., with yellow-to-orange wall that reacts K+ reddish, and its thin subcylindrical, long hairs and broadly ellipsoid to almost subspherical, smooth-walled ascospores (8.5–)9–11(12.5) × (5.2–)5.5–6.5(–7) µm.

TYPUS. SPAIN, León, Ancares, Teso de Valiñas, road LE-3203, *Quercus petraea* forest with rocks, on *Parmeliella testacea* on *Quercus petraea*, 42°51′53.1″ N, 6°51′47.1″ W, 1155–1165 m, 3–07–2021, J. Etayo 33,415 and M.E. López de Silanes (SANT-Lich. 12717–holotype designated here, hb. Etayo–isotype).

Ascomata perithecioid, lichenicolous on corticolous *Parmeliella*, firstly immersed, then erumpent, globose, densely hairy throughout the perithecia, orange when dry, without conspicuous hyphal net below the perithecia, 130–180 µm in diam., with central ostiole, not papillate. Ascomatal wall composed of several layers of cells, an internal region hyaline, composed of polygonal cells, 5–10 µm diam., and an external region yellow orangish, K+ orange red, with cells forming a textura angularis but with irregular thickening, 1–4 µm, of walls, especially thick at the angles, cells 5–13 µm diam. Setae abundant especially around ostiole, simple, cylindrical, slightly tapered to the apex, septate, thick walled, curved to vermiform, colorless, 40–60 × (2–)3–4 µm. Hymenium hyaline, with many small oil droplets. Hamathecium gelatinized. Periphyses around the ostiole abundant, 15–20 × 2–2.5(–3) µm, sometimes thickened above. Asci unitunicate, 8-spored, subcylindrical to clavate, with a thin lateral wall and a small thickening at the apex easily visible in water, less visible in K, 44–70 × 8–10 µm (n = 8). Ascospores uniseriate, rarely biseriate in the ascus, hyaline, with smooth wall, (0–)1-septate, not constricted at the septum, ellipsoid to broadly ellipsoid or almost subspherical, with obtuse ends, with a large oil droplet in each cell, (8.5–)9–11(12.5) × (5.2–)5.5–6.5(–7) µm (n = 20).

No currently known species of lichenicolous *Trichonectria* can be confused with *T. parmeliellae*. Its orange, globose perithecia covered with thin hairs, ascomatal wall orange, K+ reddish and ascospores broadly ellipsoid are very characteristic. Ascomatal wall is *Trichonectria*-type but wall thickenings are not so easily visible, so the fungus could be confused with a *Nectriopsis*. There are two species of *Nectriopsis* it could be confused with: *Nectriopsis lecanodes*, which is very common on several large lichens from the Lobarion community: *Lobaria*, *Lobarina*, *Nephroma*, *Peltigera*, and *Ricasolia* in nearby Galicia localities. This species has pinkish, large ascomata, 250–350 µm diam., hairs with thin wall, asci truncate at the end and ascospores 4–5 µm wide and spirally ornamented [59]. *Nectriopsis peruvianus* [60] has ascospores similar to those of *T. parmeliellae*. This species is known from South America growing on *Cora glabrata* and *Coccocarpia* sp. and differs from *T. parmeliellae* in its pinkish to whitish perithecia with hairs of thin wall and verruculose ascospores, similar in shape but smaller, 7–9 × 5.5–7.5 µm.

Etymology: Referring to its growth on *Parmeliella testacea*, a host with few lichenicolous fungi.

Additional specimen examined: SPAIN. Navarra, valle del Baztán, pass Vendreka, way to pass of Arregui by Orabidea, alder grove with *Castanea* sp., *Corylus avellana* and *Quercus* sp., on *Parmeliela testacea* growing on *Quercus robur*, 21 July 1993, J. Etayo 14,339 (hb. Etayo).

***Trichonectria* *rubefaciens*** (Ellis & Everh.) Diederich & Schroers ssp. ***cryptoramalinae*** Etayo & López de Silanes, ssp. nov. (Figure 5).

Mycobank MB 851494

Similar to *T. rubefaciens* but with smaller perithecia 60–120 µm diam. vs. 80–160 µm diam., smaller asci 30–37 × 5–8 µm vs. 22–45(–57) × 8–10(–12) µm, and different conidia without annular thickenings at the ends. Furthermore, it lives on species of *Ramalina* instead of Parmeliaceae family hosts.

TYPUS. SPAIN, Lugo, Courel, Mercurín, grove with old chestnut trees, on *Ramalina fastigiata* growing on *Castanea*, 42°37′58″ N 7°09′48.6″ W, 670 m, 6–July–2021, J. Etayo 33,545 and M.E. López de Silanes (SANT-Lich. 12716–holotype designated here, hb. Etayo–isotype).

Ascomata perithecioid, lichenicolous on *Ramalina* spp., superficial, doliiform or barrel-shaped, laterally compressed when dry, white yellowish or slightly pinkish to brownish, 75–100 µm high, 60–120 µm in diam., with a crown of setae on the top surrounding the ostiole. Ascomatal wall colorless to brownish, consisting of several layers, outer layer covered by a net of brownish, septate hyphae, 2–4 µm wide. Setae present only around ostiole, short and with a thick wall, (13–)17–25(–28) × 4.5–6.5 µm (10). Hymenium hyaline, KI-, when young with moniliform, broad, thin-walled interascal filaments, to 10 µm wide, composed of ellipsoid to broadly ellipsoid cells. Asci unitunicate, clavate, with obtuse end, 8-spored, 30–37 × 5–8 µm (n = 8). Ascospores subcylindrical to fusiform, straight to curved, 13–18 × 2.5–3.5 µm (n = 10). Conidial morph of *Acremonium*-type, with conidia straight, subcylindrical, hyaline, 11–15 × 2–2.5 µm, without annular thickenings in the extremes (n = 10).

*Trichonectria rubefaciens* is a common species in Europe, and it has also been recorded in the USA [61], always growing on Parmeliaceae species. According to the review of lichenicolous fungi hosts of this species by Brackel [12], it has been recorded on *Evernia*, *Flavoparmelia*, *Parmelia*, *Parmotrema*, *Platismatia*, *Pleurosticta*, and *Punctelia*. The above description of this subspecies on *Ramalina* sp. differs from the protologue only in its smaller perithecia (80–160 µm in Lowen [61]) and slightly smaller asci and spores, in addition to the different hosts. Furthermore, another Spanish sample growing on *Ramalina farinacea* also hosted its asexual morph intermixed with perithecia of *Trichonectria*, and it seems being different from the typical subspecies. Lowen [61] named this morph *Acremonium rhabdosporum*. Hawksworth [62] studied the type with conidia, which are simple elongate-cylindrical, straight, the apices with annular thickenings of the wall, 12–18 × 2–2.5 µm. Interestingly, the host of the type belongs to *Cladonia* sp. (Cladoniaceae) not Parmeliaceae. Conidia of the ssp. *cryptoramalinae* lacks those annular thickenings of the conidial wall, being another distinctive feature for differentiating from ssp. *rubefaciens*.

Etymology: The specific epithet refers to the host lichen genus (*Ramalina*) and small differences in the size of its ascomatal structures and different conidia. This is the first time a species of *Trichonectria* has been recorded on *Ramalina*.

Additional specimen examined: Gipuzkoa, Sª de Aralar, path from Lareo to Enirio, beech wood, on *Ramalina farinacea* on *Fagus sylvatica*, 775–790 m, N42°58′39″, W2°5′11″, 22 May 2020, J. Etayo 25,965 (hb. Etayo).

***Trimmatostroma*** spp.

We found at least three species belonging presumably to this genus in Galicia forests: one growing on *Phlyctis argena* that causes a grey or violaceous grey infection surrounded by dark rings. Another one was growing on *Nephromopsis chlorophylla*, which seems to be different to *T. cetrariae* [63] in its more septate conidia and not-constricted septa. Finally, there are another species growing on *Flavoparmelia*, which is very common on the Iberian Peninsula.

Loc. 8, on *Phlyctis argena* growing on *Fraxinus* sp., J. Etayo 33,359 (hb. Etayo). Loc. 3a, on *Nephromopsis chlorophylla* growing on *Quercus petraea*, J. Etayo 33,463 (hb. Etayo). 13, on *Flavoparmelia caperata* growing on *Castanea sativa*, J. Etayo 33549, 33,550 (hb. Etayo).

***Unguiculariopsis lettaui*** (Grumm.) Coppins

This species is very common on *Evernia prunastri* in Spain. In the studied area, most of their thalli suffers yellow to brownish infections caused by this and other species like *Lichenoconium erodens*, *Licea parasitica*, or *Myxotrichum bicolor*.

Loc. 0, on *Evernia prunastri* on *Prunus* sp., M.E. López de Silanes (SANT-Lich. 12465). Loc. 1, on *E. prunastri* growing on *Castanea sativa*, J. Etayo 33376, 33,789 (SANT-Lich. 12574, 12575). Loc. 2b, on *E. prunastri* growing on twigs, J. Etayo 33,582 (hb. Etayo). Loc. 3a, on *E. prunastri* growing on *Quercus petraea*, J. Etayo 33,422 (SANT-Lich. 12598). Loc. 8, on *E. prunastri* growing on *Corylus avellana*, J. Etayo 33,364 (hb. Etayo). Loc. 9, on *E. prunastri* growing on *Fagus sylvatica*, J. Etayo 33,629 (hb. Etayo).

***Unguiculariopsis thallophila*** (P. Karst.) W.Y. Zhuang

This species lives commensally on different species of *Lecanora*, mainly corticolous ones like *L. carpinea*, *L. chlarotera* aggr., *L. leptyrodes*, *L. septentrionalis* and *L. subcarnea* [52].

Loc. 0, on *Lecanora chlarotera* growing on twigs of *Acer* sp., J. Etayo 33,460 (SANT-Lich. 12626). Loc. 8, on *L. carpinea* growing on *Corylus avellana*, J. Etayo 33,362 (hb. Etayo).

***Vouauxiella lichenicola*** (Linds.) Petr. & Syd.

Loc. 12, on *Lecanora chlarotera* growing on twigs of *Juglans regia*, J. Etayo 33,553 (SANT-Lich. 12608). Ibidem, J. Etayo 33,562 (hb. Etayo).

***Xanthoriicola physciae*** (Kalchbr.) D. Hawksw.

Loc. 7, on thallus and apothecia of *Xanthoria parietina* growing on *Sambucus nigra*, J. Etayo 33,471 (hb. Etayo).

***Xenonectriella fissuriprodiens*** (Etayo) Etayo & van den Boom

Originally described by Etayo and Diederich [43], we found it only on unhealthy, bleached zones of *Lobaria pulmonaria* in two Galician localities.

Loc. 9, on *L. pulmonaria* growing on *Fagus sylvatica*, J. Etayo 33,622 (SANT-Lich. 12613). Loc. 12, on *L. pulmonaria* growing on *Castanea sativa*, J. Etayo 33,558 (hb. Etayo). Ibidem, J. Etayo 33,564 (hb. Etayo).

***Xenonectriella*** cf. ***lutescens*** (Arnold) Weese

Diagnostic features of the Galician material are: ascomata immersed, with only the ostiole emerging above the host cortex; asci with only 1 composite ascospore since very young; ascospores firstly hyaline then brown, submuriform to muriform produced by fusion of eight spores, ornamented with a thin reticulum, 58–84 × 14–17 µm. The infection on *Sticta* gives a purplish hue to the host hyphae.

*Xenonectriella lutescens* (Rehm.) Weese has been recorded in central Europa, Russia, and North America on *Collema fasciculare*, *Peltigera*, *Solorina bispora*, *S. saccata* (type), and *Peltigera* [12,47,64,65] but never on *Sticta*. Regarding Zhurbenko [47], fused ascospores are (19–)22–38(–65) × (7–)9–11(–13) µm, (4–)2(–1) per ascus. According to Rossmann et al. [66], ascospores are much smaller, 23–32 × 7.5–11 µm. Our spores are notably larger than those studied by these authors, which could indicate that the name *X. lutescens* references a group of different, related species. Molecular analyses would be necessary to solve the dilemma, but it is a rare group of scarcely collected species.

Loc. 11, on *Sticta fuliginosa* growing on *Castanea sativa*, J. Etayo 33489 (hb. Etayo).

***Xenonectriella septempseptata*** (Etayo) Etayo & van den Boom

This species was described by Etayo [67] as growing on *Melanelixia glabratula.* It is easily recognizable by its long and multiseptate ascospores, (3–5–)7-septate, 41–63 × 4.5–6 µm. Galician samples have spores that are a bit larger, (3–)5(–9)-septate, 55–73 × 5.5–7 µm.

Loc. 8, on *Melanelixia glabratula* growing on *Ilex aquifolium* and *Fagus sylvatica*, J. Etayo 33,357 (hb. Etayo). Loc. 12, on *M. glabratula* growing on *Castanea sativa*, J. Etayo 33,557 (hb. Etayo).

***Zyzygomyces bachmannii*** (Diederich & M.S. Christ.) Diederich, Millanes & Wedin

≡ *Syzygospora bachmannii* Diederich & M.S. Christ.

This is a common species in the study area, forming brown tuberculate galls on healthy squamules and podetia of *Cladonia rangiformis* and *C. parasitica*, the latter of which was not recorded as a host by Diederich [68] or Diederich et al. [69].

Loc. 1, on squamules and podetia of *Cladonia parasitica* growing on *Castanea sativa* wood J. Etayo 33,383 (hb. Etayo). Loc. 10, on *C. rangiformis* growing on limestone, J. Etayo 33,660 (hb. Etayo, SANT-Lich. 12619).

***Zyzygomyces physciacearum*** (Diederich) Diederich, Millanes & Wedin

≡ Syzygospora physciacearum Diederich

Basidiomata of this species growing on *Physcia leptalea* are here pure white, not pale to dark brown, as stated in Diederich et al. [68].

Loc. 10, on *P. leptalea* growing on *Acer* sp. and on *Physcia aipolia* growing on *Prunus* sp., J. Etayo 33,653 (SANT-Lich. 12620). Loc. 12, on *P. leptalea* growing on twigs of *Juglans*, J. Etayo 33,554 (hb. Etayo).

## Figures and Tables

**Figure 1 jof-10-00060-f001:**
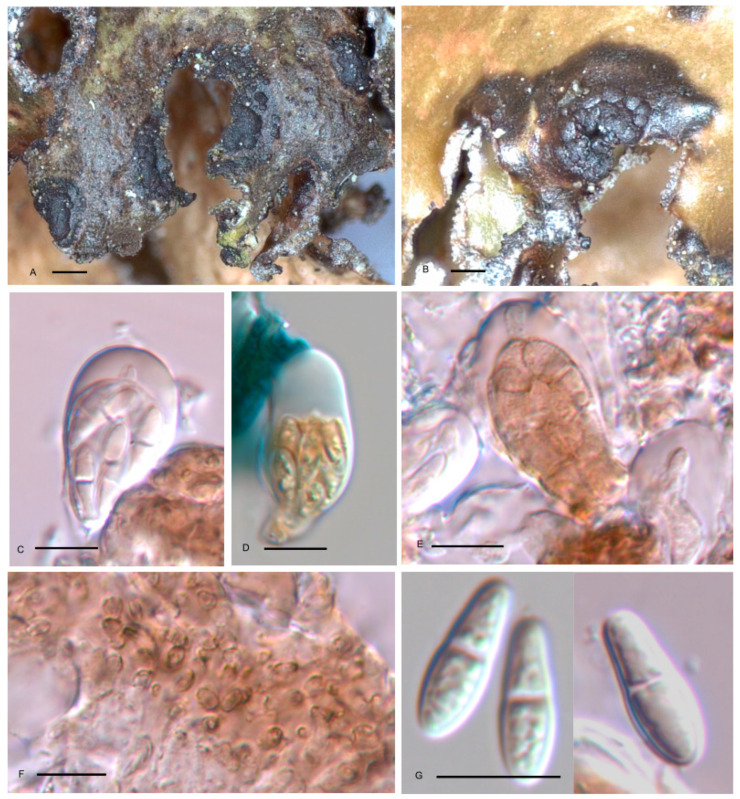
*Arthonia boomiana* (holotype). (**A**,**B**) Apothecia of the fungus on lobes of *Nephromopsis chlorophylla*. (**C**) Ascus (water). (**D**) Ascus (KI). (**E**) Ascus with brown spores (water). (**F**) Epihymenium with top cells of paraphysoids. (**G**) Ascospores. Scales: (**A**,**B**) = 250 µm, (**C**–**G**) = 10 µm.

**Figure 2 jof-10-00060-f002:**
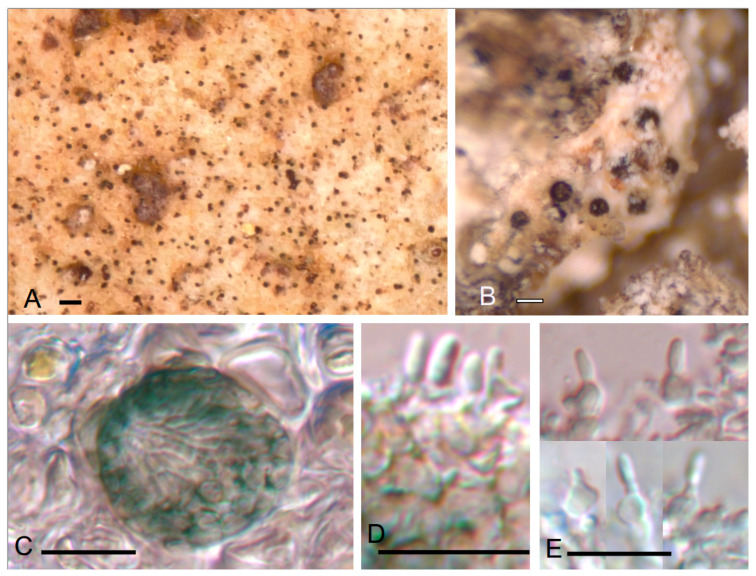
*Lawalreea burgaziana* (holotype). (**A**) Pycnidia on the surface of *Platismatia glauca*. (**B**) *Briancoppinsia cytospora* on *Lepra albescens* (hb. Etayo 25311) showing its much larger pycnidia. (**C**) A pycnidium in surface view (**D**) Paraplectenchymatous pycnidial wall. (**E**) Conidiogenous cells and conidia (in K). Scales: (**A**,**B**) = 100 µm, (**C**–**E**) = 10 µm.

**Figure 3 jof-10-00060-f003:**
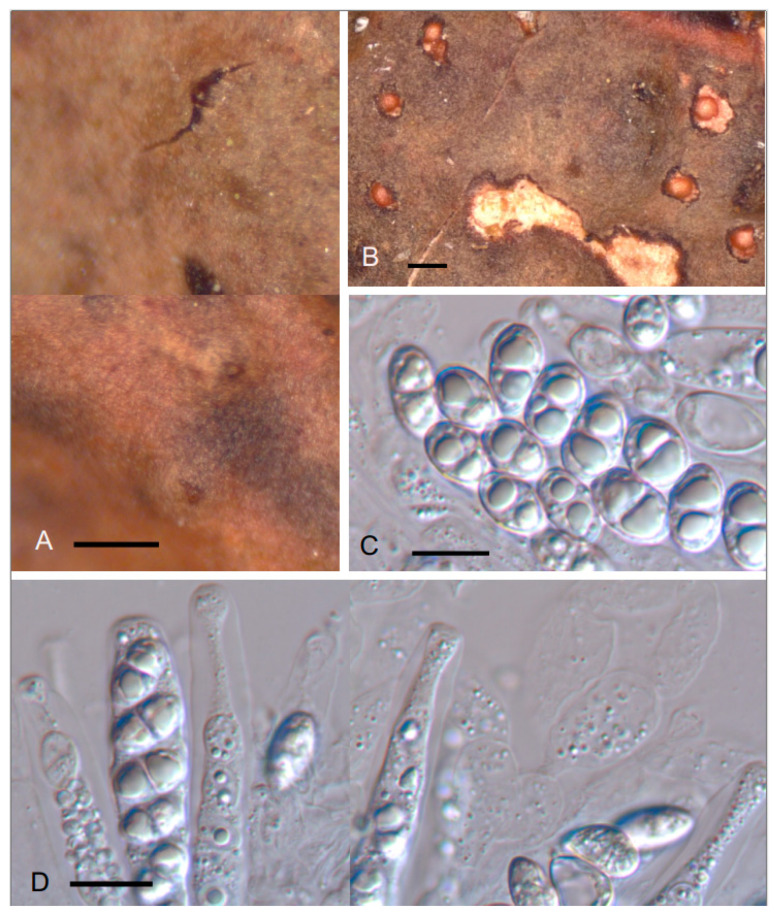
*Pronectria scrobiculatae* (holotype). (**A**) Ascomata immersed in the *Lobarina scrobiculata* thallus, only emerging a small papillae (**below**) or breaking finally the thallus (**above**). (**B**) Holes left by perithecia on the host thallus. (**C**) Ascospores in water. (**D**) Upper part of asci and paraphyses in water. Scales: (**A**,**B**) = 250 µm; (**C**,**D**) = 10 µm.

**Figure 4 jof-10-00060-f004:**
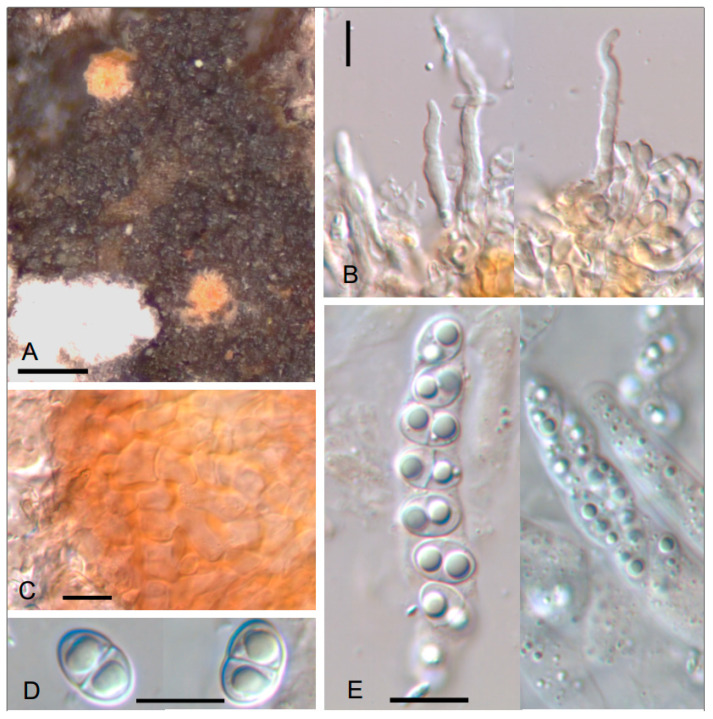
*Trichonectria parmeliellae* (holotype). (**A**) Hairy perithecia growing on *Parmeliella testacea*. (**B**) Ascomatal hairs with a thick wall (in K). (**C**) Ascomatal wall in K turning reddish (lower part). (**D**) Ascospores in water. (**E**) Asci with uni- and biseriate spores (in K). Scales (**A**) = 250 µm, (**B**–**E**) = 10 µm.

**Figure 5 jof-10-00060-f005:**
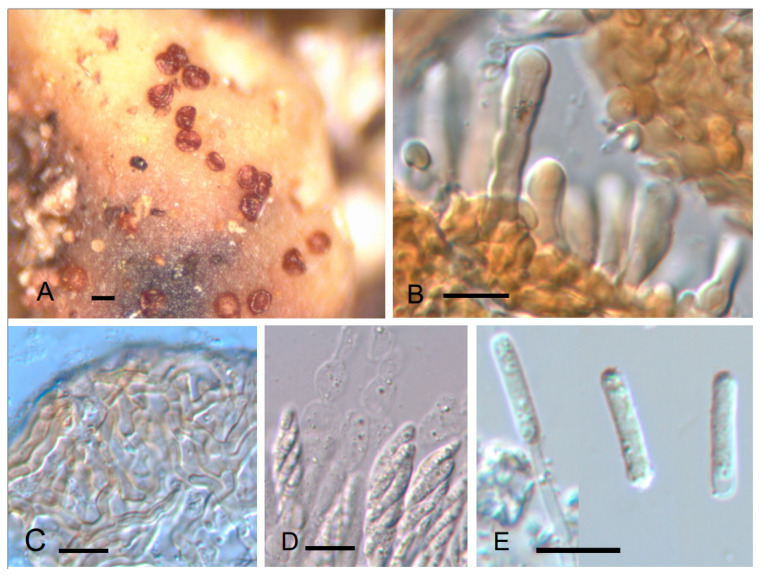
*Trichonectria rubefaciens* ssp. *cryptoramalinae* (holotype). (**A**) Ascomata on the apothecial disc of *Ramalina fastigiata*. (**B**) Ascomatal hairs. (**C**) Ascomatal wall covered by a net of hyphae. (**D**) asci and paraphyses. (**E**) Conidia without apical widenings. Scales: (**A**) = 100 µm; (**B**–**E**) = 10 µm.

## Data Availability

The material studied has been deposited in a private herbarium, herb. J. Etayo, and in a public herbarium, SANT. Therefore, it is possible to consult and review these samples.

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
