# Peer review of "Contribution to the Study of Lichenicolous Fungi from Northwest Iberian Peninsula (León and Lugo Provinces)"

_jof, 2024, doi:10.3390/jof10010060_

Round 1
Reviewer 1 Report
Comments and Suggestions for Authors
The work was carried out by highly qualified and experienced specialists in their field and makes a significant contribution to the knowledge of the lichenicolous mycobiota of Europe, including the description of a number of new species/subspecies. I have no doubts about the quality of the manuscript, but I have suggested numerous corrections in the form of presentation of the material and especially in the English language, which I have noted directly in the manuscript.

Although I have offered numerous English corrections, I am not a native English speaker and therefore suggest to show the manuscript, after taking my corrections into account, again to a native English speaker.
Author Response
Answers to M.Z. review (1).
Thank you for your contributions and corrections. I have used almost all of them.
The problem with the title is not easy to solve. It is generally accepted that you can consider other fungi like lichen-forming fungi, bacteria or myxo (protozoa) as part of a paper entitled with "lichenicolous fungi" words. As an example the last book of Lichenicolous fungi of the world. BAsidiomycetes, also deals with gall-forming bacteria. We must aknowledge that anybody would like a title like: Contribution to the Study of Lichenicolous Fungi and lichenicolous lichens, bacteria and protozoa from NW Iberian Peninsula (León and Lugo provinces).
I have made a note about that fact in the introduction chapter.
- If the parasite grows on the host thallus nothing is said but we put on apothecia or hymenium when necessary.
- Today the current name is Umbilicaria pustulata.
The proofreader suggests shortening the list if there is only one host with a previous comment about the host. However, we think it is better to maintain the homogeneity of the text.
Many thanks for your consideration and time dedicated to the ms!
Javier Etayo
Reviewer 2 Report
Comments and Suggestions for Authors
Dear authors,
I am very pleased to see such a relevant contribution to the knowledge of lichenicolous fungi.
In my opinion, the manuscript is well structured, and the new taxa are properly described and commented.
I am asking for a minor revision given that:
a) the English do require a revision by a native English speaker, and
b) the introduction, given that not all the readers of Journal of Fungi are lichenologists, should provide at least in a few lines a description of what a lichenicolous fungus is.
All the best
Comments on the Quality of English LanguageAs stated before, please ask to a native English speaker to review the manuscript
Author Response
Thank you very much for your kind report on our ms,
I have incorporated your suggestions, particularly regarding the introduction of lichenicolous fungi organisms.
Best regards
Reviewer 3 Report
Comments and Suggestions for Authors
The manuscript is a list of lichen-inhabiting species found from a region of Spain. It is a pity to say that the manuscript is full of typing errors, unfinished sentences, and in some places, there are Spanish words instead of English, the geographical coordinates and dates should be in the same format throughout the text. If the host species in mentioned several times in the list, then only when first mentioned, the name should be fully written out. In some places it is difficult to understand the descriptions. I marked all the errors that I found in yellow. These sentences, problematic places, etc. which are unclear to me are marked in green. I really hope that this is not draft that was sent to me. The language of the manuscript needs correction.
· What I miss the most is the map of the country, and a map of the study region with spotted localities.
· In the Introduction you mention that this region was visited several times, but was it a part of larger project or was it a project by itself?
L. 113.-114. is it correct that it was found on Melanelixia ?
L. 116 – There are several species which spores are 3-septate. More details are needed.
L. 123 – are the conidia indeed brown?
L. 132 – do I understand correctly that the conidiophores are septate? Are these structures pigmented?
L. 159 – What you mean “Our taxon looks like this species by its …” If you are not certain, add cf. or aff.
L. 161 – what are thinner? ascospores? septum?
L. 167 – in the diagnosis, you should clearly say how your new species differs from the rest of lichenicolous species of this genus; you also compare your new species with A. xanthoparmeliarum, but in the text you compare with many other species. Why so ?
L. 188 and 190 – what means numbers 8 and 26 in brackets?
L. 241 – Why Catillaria between quotation marks?
L: 247 – I do not understand this sentence: “… except for the smaller size in asci.”
L. 247-248: also, this sentence is confusing – “Asci on KI are also Catillaria-type on Smith et al. [16].”
L. 307 – what is 22 in brackets?
L. 322 – it seems that the studied specimens are so variable that it is difficult to believe that they represent the same species. Maybe it is better to add “cf.”
L. 339 – the ascospores are longer compared with what species?
L: 348 – what you mean by saying “old named”
L. 352 – there are several species that form galls on Cladonia thalli.
L: 374-376 – complicate to understand. Are the asexual stages grow on Hemigrapha species?
L. 379-381 – How could it confirm your observation?
L: 386 – what the abbreviation “EEUU” mean?
L. 420 – I do not understand this sentence “…then conidia and conidiogenous cells rests that furthermore.”
L: 423 – I do not understand the position of conidiogenous cells?
L: 607 – should be two separate sentences, otherwise it is confusing.
L: 613 – in which country are La Rioja and Soria ?
L. 662 – what means “… forming …. galls in one locality (10)
L. 671 – why it is important to mention that “… we found this species …”
L. 724 – if you say that your new species is the most similar to Pronectria robergei, then you have to say in which aspects these two are similar.
L. 751 – I do not understand.
L. 792 – On Trentepohlia thallus?
L. 870 – Index Fungorum does not give Peltigera furfuracea
L. 876 – something missing in this sentence.
L. 887 – why this sentence is necessary?
L. 920 – does this mean that in Trichonectria species are not orange?
L. 999 – there is no such variety as var. typica. Should be var. rubefaciens instead.
L. 1076 – could be the species on Physcia aipolia, Zyzygospora aipoliae?
L. 1102 – Why you say that “The discussion is included …” It is unnecessary.

The language is poor. Needs extensive correction.
Author Response
Answers to the issues proposed by A.S. (review 1).
Thank you for your contributions and corrections. I have used almost all of them.
113.-114. is it correct that it was found on Melanelixia ? corrected
116 – There are several species which spores are 3-septate. More details are needed. Yes, but only one with this feature and growing on Ramalina
123 – are the conidia indeed brown? Yes
132 – do I understand correctly that the conidiophores are septate? Are these structures pigmented? mistake, conidiogenous cells, hyaline
159 – What you mean “Our taxon looks like this species by its …” If you are not certain, add cf. or aff. yes, changed.
161 – what are thinner? ascospores? septum? ascospores
167 – in the diagnosis, you should clearly say how your new species differs from the rest of lichenicolous species of this genus; you also compare your new species with A. xanthoparmeliarum, but in the text you compare with many other species. Why so ? In a diagnosis you can not compare with all the species of the genus, in the case of lichenicolous Arthonia more than 50. You must compare with the one is more similar to the new species, in this case A. xanthoparmeliarum.
188 and 190 – what means numbers 8 and 26 in brackets?
241 – Why Catillaria between quotation marks? OK, firstly, it was not considered a Catillaria.
L: 247 – I do not understand this sentence: “… except for the smaller size in asci.” Yes, you are right it was understandable. Elliminated.
247-248: also, this sentence is confusing – “Asci on KI are also Catillaria-type on Smith et al. [16].” Right. Elliminated.
307 – what is 22 in brackets? number of measuring spores.
322 – it seems that the studied specimens are so variable that it is difficult to believe that they represent the same species. Maybe it is better to add “cf.” Yes, I have changed the sentence to avoid doubts.
339 – the ascospores are longer compared with what species? yes, corrected.
L: 348 – what you mean by saying “old named” It was only recorded on Lasallia, now considered Umbilicaria.
352 – there are several species that form galls on Cladonia thalli. Yes, this is one of them.
L: 374-376 – complicate to understand. Are the asexual stages grow on Hemigrapha species? No. We found only conidiomata of that species of Hemigrapha, not ascomata. I will explain a little better.
379-381 – How could it confirm your observation? It seems than this species does not form teleomorph in Europe, but you are right it is a bit uncorrect.I am changing the sentence.
L: 386 – what the abbreviation “EEUU” mean? Sorry, corrected (USA).
420 – I do not understand this sentence “…then conidia and conidiogenous cells rests that furthermore.” Yes, changed.
L: 423 – I do not understand the position of conidiogenous cells?
L: 607 – should be two separate sentences, otherwise it is confusing. Yes, changed.
L: 613 – in which country are La Rioja and Soria ? changed.
662 – what means “… forming …. galls in one locality (10) Bad written. Corrected.
671 – why it is important to mention that “… we found this species …” Yes, you are right. Changed.
724 – if you say that your new species is the most similar to Pronectria robergei, then you have to say in which aspects these two are similar. But this is a diagnosis, we have to highlight the differences.
751 – I do not understand. Changed.
792 – On Trentepohlia thallus? changed
870 – Index Fungorum does not give Peltigera furfuracea, lapsus: membranacea
876 – something missing in this sentence. Rewritten
887 – why this sentence is necessary? It is not strictly necessary but it is not common to find it growing on lichens.
920 – does this mean that in Trichonectria species are not orange? Not, it differs by the whole features that are exposed in the diagnosis, but also for the color: most of Trichonectria on Usnea are dark brown even almost blackish.
999 – there is no such variety as var. typica. Should be var. rubefaciens instead. Yes, lapsus.
1076 – could be the species on Physcia aipolia, Zyzygospora aipoliae? Do you mean Zyzygomyces? I have seen it and I think it is Z. physciacearum but...
1102 – Why you say that “The discussion is included …” It is unnecessary. Yes, it was a note for the redaction.
Thanks a lot for your interesting commentaries.
Javier
Round 2
Reviewer 1 Report
Comments and Suggestions for Authors
In my opinion the authors have taken into account the suggestions I made to improve the manuscript during the initial review . It seems to me that everything is now in order.
Author Response
Many thanks for your reports about the paper entitled: "Contribution to the study of the lichenicolous fungi from NW Iberian Peninsula (León and Lugo provinces).
Javier ETayo
Reviewer 3 Report
Comments and Suggestions for Authors
The manuscript has improved a lot, but there are still a lot of typos, etc. I marked these in yellow in pdf. In several places dots are missing at the end of sentences; there are inconsistencies how the information about examined specimens is given. For me it is not clear the formula how measurements are presented. There are probably minimum and maximum values in brackets but what are the numbers outside of brackets. Also, sometimes number of measurements is given, sometimes not. Please specify that in Materials and methods. In the list of references, some journal names are abbreviated, some are not. Please unify.
There are also some more specific questions.
1. Lines 51-52: In the line 51, you give a formula for KOH + abbreviation while in line 52 you use the name. Should be that the name is in the line 51 and in brackets formula and abbreviation.
2. L. 53-54 – Please specify in which cases you used Congo red and Cresyl blue.
3. L. 134 – in most of the literature sources, the conidia of Abrothallus usneae are described as being hyaline. Are you sure that if they are brown, they belong to A. usneae at all ? There may be other Abrothallus species with brown conidia.
4. L. 212 – You said: “Its habitus could like Arthonia epiphyscia”. Why “could”? And is this sentence necessary?
5. L. 256 - The sentence “We do not find asci or spores inside” should be in past tense.
6. L. 261 – From the description I read that the species may be not even close to Catillaria lobariicola. But then you can’t call it Catillaria lobaricola even in between quotation marks.
7. L. 280 – I’m confused. You said that “Hafellner [22] found it, like we did, on species of Parmelina (P. carporhizans and P. quercina).” But your record is on Parmelina pastellifera ?.
8. L. 322 – should be “becomes”
9. L: 383 – Why it is necessary to say that “the old named Lasallia (=(now synonym of Umbilicaria)” ?
10. L. 388 – There are several species on Cladonia that form galls on the thallus. Please specify or leave this sentence out.
11. L. 414-415: You say that “since the sizes of the conidiomata and conidia in our specimen are very similar to …”. It will be easier to follow, compare and accept if you give measurements of the original description for comparison.
12. L. 446 – should be “width”.
13. L. 469 – it could be not “principally” if it has been recorded also on Cladonia, etc.
14. L: 485 – What is “… regarding Ryan [36]…”
15. L: 489 – should be “open”
16. L. 592 – In what respect this species is unusual ?
17. L: 630-631 – This sentence “Two sterile, unidentified species of this genus were found growing on saxicolous rocks” does not carry any information if you do not give any more information which characteristics say you that these two species are undescribed
18. L. 661 – You say that “… fit well with the previous description on other species”. This is not enough. Please give at least some measurements that would create certainty that your determination is correct, and the measurements fit the description.
19. L. 663 – Should be “On Parmelia sulcata it causes …
20. L. 682 – The sentence “Lichenized fungus living typically on saxicolous lichens like Xanthoparmelia in Galicia” does not carry much information if you do not say how it looks like or give a reference for the description.
21. L: 723 – there are several species that may form galls on Hypogymnia. So, either delete this sentence or specify.
22. L. 741 – Is this sentence important?
23. L. 759-760 – I do not understand the sentence “We also found the host on trees with no presence of the fungus”.
24. L: 882 – What is the K reaction? The sentence seems to be incomplete
25. L: 1059 – I still do not fully understand the reasoning to describe it as subspecies? What are the limits and differences between species and subspecies? It would be better to treat it as species until molecular data will be available.

Author Response
Dear reviewer,
I have corrected almost all your references to improve the manuscript. I am attaching a Word document with answers to all your improvements.
Thanks a lot and best regards
